# ADuLT: An efficient and robust time-to-event GWAS

Emil M. Pedersen [1,2] ✉, Esben Agerbo[1,2,3], Oleguer Plana-Ripoll[1,4], Jette Steinbach [1], Morten D. Krebs[5], David M. Hougaard [6], Thomas Werge [5,7,8], Merete Nordentoft[2,9], Anders D. Børglum [2,10,11], Katherine L. Musliner[1,12,13], Andrea Ganna [14], Andrew J. Schork[5,8,15], Preben B. Mortensen [1,2], John J. McGrath [1,16,17], Florian Privé[1,2,20] & Bjarni J. Vilhjálmsson [1,2,18,19,20] ✉

Proportional hazards models have been proposed to analyse time-to-event phenotypes in genome-wide association studies (GWAS). However, little is known about the ability of proportional hazards models to identify genetic associations under different generative models and when ascertainment is present. Here we propose the age-dependent liability threshold (ADuLT) model as an alternative to a Cox regression based GWAS, here represented by SPACox. We compare ADuLT, SPACox, and standard case-control GWAS in simulations under two generative models and with varying degrees of ascertainment as well as in the iPSYCH cohort. We find Cox regression GWAS to be underpowered when cases are strongly ascertained (cases are oversampled by a factor 5), regardless of the generative model used. ADuLT is robust to ascertainment in all simulated scenarios. Then, we analyse four psychiatric disorders in iPSYCH, ADHD, Autism, Depression, and Schizophrenia, with a strong case-ascertainment. Across these psychiatric disorders, ADuLT identifies 20 independent genome-wide significant associations, case-control GWAS finds 17, and SPACox finds 8, which is consistent with simulation results. As more genetic data are being linked to electronic health records, robust GWAS methods that can make use of age-of-onset information will help increase power in analyses for common health outcomes.

Over the last decade, genome-wide association studies (GWAS) have successfully identified thousands of genetic variants associated with human diseases[1,2]. Most of these GWASs have modelled the outcome as a binary case-control variable in a logistic (or linear) regression while accounting for covariates such as age, sex, and genetic principal components. However, these models are generally not suited for modelling time-to-event data, as they do not account for certain types of missing or censored data. Time-to-event models are commonly used in epidemiology and many other fields, and have proven useful for both accounting for censoring, changes in disease incidence over time (cohort effects), and age-of-onset[3]. Time-to-event models can also be used to estimate absolute time-dependent risk (i.e. the probability of developing the disease as a function of time) conditional on individual features, and are therefore widely used to estimate disease risk in clinical settings[4].

Although time-to-event models have been proposed for GWAS[5–8], their adoption has been limited in practice. One reason is that age-of-onset (AOO) information is often not made available. However, time-to-event data is becoming more readily available as more genotyped data are being linked to health records. Another reason is that fitting

these models on large data is computationally intensive. However, several computationally efficient survival analysis GWAS methods have been proposed recently for large population-scale data. These include efficient Cox regression implementations[9,10], and an efficient frailty (random effects) model[11]. The frailty model inherits some of its advantages from the mixed model[12–15], and can both account for population structure and relatedness, as well as improve statistical power when sample sizes are large. A third reason is that time-to-event models are generally not expected to provide significant gains in power for rare health outcomes[16]. Indeed, the performance of Cox-based regressions in a GWAS setting is poorly understood, and they have only been viewed in comparison to other Cox-based regressions or logistic regression[5,8]. Importantly, these benchmarks have focused on the proportional hazards generative model and without significant case ascertainment, which is common in GWAS. In practice, when collecting data for GWAS it is common to oversample cases to increase the effective sample size and statistical power in the genetic analyses, leading to a case-control or case-cohort study design.

Here we examine to what extent case ascertainment in GWAS data affects Cox regression GWAS and standard case-control GWAS. Inspired by how robust liability threshold models[17,18] (LTM) have proved to be for ascertained data[19], we propose ADuLT (age-dependent liability threshold) as a computationally efficient time-to-event model for GWAS, and examine how it performs in the presence of case ascertainment. ADuLT is based on the liability threshold model and is the underlying model for the recently proposed LT-FH++ method[20]. ADuLT accounts for age-of-onset information, as well as sex and cohort effects by personalising the thresholds used to infer the case-control status for each individual. These thresholds are personalised by using population-based cumulative incidence proportions (CIPs) for the phenotype of interest as a function of age and additional information, such as sex and birth year (to model sex and cohort effects). We examine how ADuLT compares to SPACox and standard linear regression GWAS in terms of both statistical power and computational efficiency, using both simulations and real iPSYCH data, which is a psychiatric disorder case-cohort data with a strong case ascertainment bias where cases are about 20 times more likely to be sampled[21,22].

With an increasing integration between biobanks and electronic health records, it is important to utilise additional information in the best way possible, and we believe that knowledge about age-of-onset will be a common and powerful piece of information to include. Finally, ADuLT is implemented in an efficient R package called LTFHPlus (github.com/EmilMiP/LTFHPlus), and is made highly scalable by relying on parallelization and the R package Rcpp, which offers a seamless integration of R and C++[23].

## Results

### Overview of method

The age-dependent liability threshold model presented here was first introduced in our previous paper extending the LT-FH method to account for family history as well as age-of-onset, sex, and cohort effects among all individuals, including the family members[20,24]. In this paper, we focus on the ADuLT model as an alternative to commonly used time-to-event or linear regression GWAS methods, without considering any family history.

The ADuLT model modifies the LTM by assuming that the threshold used to determine an individual's case-control status corresponds to the CIP at the age of diagnosis. Only individuals with liabilities above their assigned liability threshold, which depends on their age, sex, and birth year, become a case. In Fig. 1, we present the CIPs for ADHD for individuals born in Denmark in the year 2000. The CIPs increase as the population gets older, which in turn leads to a decreased threshold. If additional information, such as sex and birth year, is available, the population CIPs should be stratified according to this additional information (as seen in Fig. 1), as this improves estimation of the genetic liability[20]. In the first step, a personalised threshold is assigned to each individual based on their current age or the age-of-onset, as well as sex and birth year. In the second step, the ADuLT model uses the liability-scale heritability to estimate a genetic liability for each individual. The third step uses the ADuLT phenotype as a continuous outcome in a GWAS. There are no restrictions on the choice of GWAS method as long as it accepts continuous outcomes, allowing researchers to benefit from current and future advances in GWAS methods. Note that Fig. 1 illustrates the use of CIP for cases. If an individual is a control, the area of possible liabilities

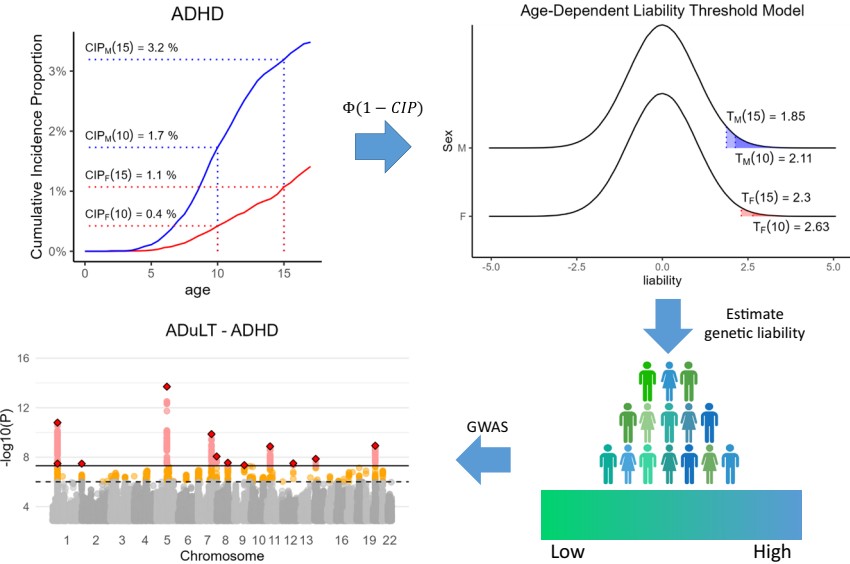

**Fig. 1 | Overview of the information used, and the different steps needed to perform a GWAS based on the ADuLT phenotype.** The cumulative incidence proportions (CIPs) stratified by sex and birth year (here ADHD for individuals born in Denmark in 2000) are converted to a threshold for the age-dependent liability threshold model. Females are represented by the red line, while males are represented by the blue line. The CIPs has been marked at the age of 10 and 15 for both sexes (dotted lines). Finally, a genetic liability is estimated for each individual, and this ADuLT phenotype can be used as the outcome in a GWAS. Parts of the plot were created with BioRender.com.

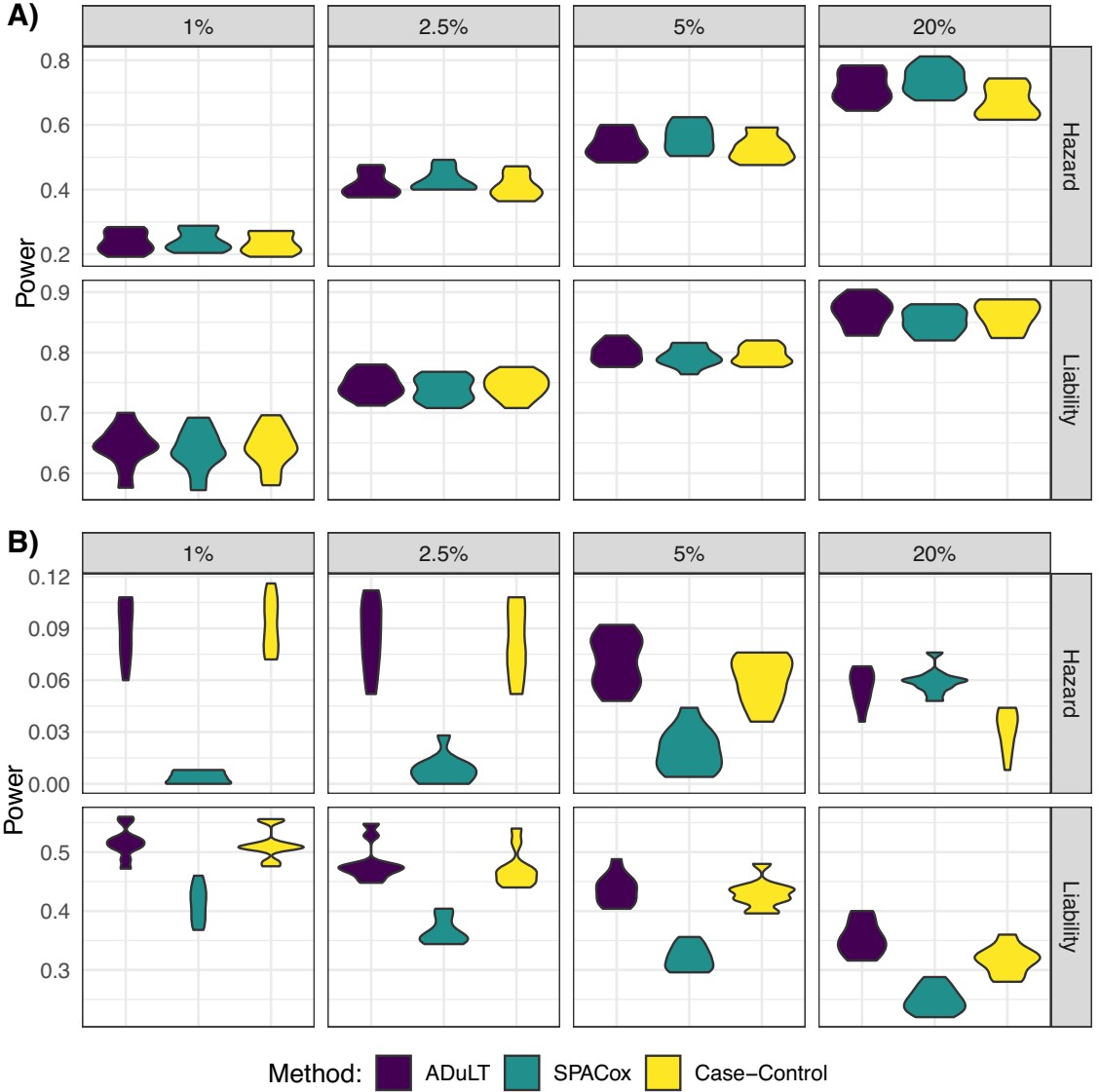

**Fig. 2 | The power is shown for different population prevalence, varying from 1% to 20%.** The generative model for `Hazard` is the proportional hazards model, and for `Liability` it is the liability threshold model. The simulation results are based on 10 replications. **A** The power, i.e. the fraction of causal SNPs detected for each method, without case ascertainment. **B** The power with case ascertainment, i.e. the number of individuals is subsampled to 10k cases and 10k controls.

will instead be from negative infinity to the threshold identified from the CIPs.

## Simulation results

We used two generative models for the simulations, namely the LTM and the proportional hazards model (see Methods). The performance of a simple case-control GWAS, SPACox, and the ADuLT phenotype used as the outcome in a linear regression-based GWAS was assessed under both generative models. Sex and age or age-of-onset were simulated for 1 million individuals, each with 20,000 independent SNPs. To examine the effect of ascertainment of cases, which is common in GWAS data, similar analyses were performed where the total number of individuals was randomly downsampling from 1 million to 20,000 individuals, leaving 10,000 controls and 10,000 cases in each case ascertained dataset. The following simulation results are based on 10 replications of each parameter setup.

Figure 2 displays the power for each method under both generative models with 250 causal SNPs. A similar plot showing the power of the same generative models but with 1000 causal SNPs can be found in Supplementary Fig. 5. Without case ascertainment, the power of all

three methods is similar under both generative models (Fig. 2A). In Fig. 2B, which is based on a case ascertained dataset, the power of all three methods decreased due to a reduced sample size, but the power of SPACox was disproportionately affected by case ascertainment. For simulated case ascertained traits that have a lifetime prevalence of 5% or below, SPACox performs worse than linear regression for both the case-control status and the ADuLT phenotype by more than a factor of 10 in the worst case, and ~25% worse in the best case. Under the proportional hazards model and a lifetime prevalence of 20%, and with case ascertainment, SPACox has an average power on par with ADuLT. The full simulation results can be found in Supplementary Data 1, with all simulation results under both generative models and with and without case ascertainment.

As several phenotypes are assigned to the same set of genotypes, we will consider a paired power for comparison in the following sections. This means all calculations are done within each iteration when possible, as the naïve mean value would not be able to account for one method consistently identifying more SNPs. Without case ascertainment, for both 250 and 1000 causal SNPs, and all considered population prevalences, the power of all methods are within ~3% of one

Run time of ADuLT & SPACox

**Fig. 3 | Each point represents the mean value of 10 replications, while the error bars are represented by the estimate ± 1.96 standard errors.** Run times were assessed for a varying number of individuals and SNPs. The number of SNPs (M) varies from 100k to 1 million, while the number of individuals vary from 10k to 100k.

SPACox uses a single CPU core, as no parallelization is available. We used 1, 2, and 4 CPU cores for estimating the ADuLT phenotype and performing the linear regression GWAS for this phenotype. The means and corresponding standard errors of the run times can be found in Supplementary Data 1.

another, except for one parameter setup. The parameter setup with the largest difference without case ascertainment was under the proportional hazards model and with 1000 causal SNPs, where the ADuLT phenotype had ~9% lower power and case-control status was 19% lower. Under the proportional hazards model and without case ascertainment, SPACox had the highest power for all considered prevalences, while case-control status had the lowest. Under the LTM and without case ascertainment, ADuLT obtained the highest power for all prevalences, while SPACox had the lowest power.

Under the LTM, with case ascertainment, and 1000 causal SNPs, the average increase in power was 117% with ADuLT over SPACox across all prevalences considered, and it was 97% for case-control status over SPACox. With case ascertainment and 250 causal SNPs, we observed an average increase of 34% in power over SPACox with ADuLT, and a 29% increase in power with case-control status, showing that SPACox has a comparatively low power for low effect sizes.

Under the proportional hazards model and with case ascertainment, we were not able to use a paired power for comparisons, as SPACox was unable to identify any genome-wide significant SNPs in 36 out of 80 simulations (29 out of 40 when simulating 1000 causal SNPs, and 7 out of 40 when simulating 250 causal SNPs). The ADuLT phenotype and case-control status were unable to identify genome-wide significant SNPs in only 6 out of 80 simulations, all of which are for 1000 causal SNPs. The following comparisons are based on the average un-paired power. Under the proportional hazards model simulations with case ascertainment and 1000 causal SNPs, ADuLT had a 317% higher power than SPACox, whereas case-control status had 256% higher power than SPACox. When simulating 250 causal

SNPs, ADuLT resulted in a 234% higher power and case-control status had a 193% higher power compared to SPACox. Plots of the power as a function of MAF and the true effect size can be found in Supplementary Figs. 8–15.

In Supplementary Figs. 6 and 7, the average $\chi^2$-statistics for the null SNPs is reported. The null SNPs are the SNPs with an effect size of 0 (i.e. no effect). The expected average of these SNPs' $\chi^2$-statistics is 1. Plots were achieved for 250 and 1000 causal SNPs, respectively, and each plot contains results for four different lifetime prevalences, with and without case ascertainment, and for both generative models. All models were well calibrated, since no inflation of the null statistics is observed. All methods controlled for type-1 errors (false positives) at varying significance levels. No false positives were observed for any method with a significance level of $5 \times 10^{-8}$. All false positive results can be found in Supplementary Data 1.

**Computation times**

The computational time for estimating the ADuLT phenotype depends solely on the number of individuals. The running time for the GWAS step depends heavily on the implementation of the GWAS method used. In Fig. 3, the combined running times of estimating the ADuLT phenotype and performing a GWAS using the bigsnpr package[25] are reported. We used 4 CPU cores for both steps, which is a conservative number of cores. The SPACox implementation does not support parallelization, which is why SPACox was run sequentially. We find that ADuLT together with a linear regression is faster than SPACox, even with only modest parallelization. Logistic regression of a binary phenotype is slower than linear regression of the same phenotype[25], which

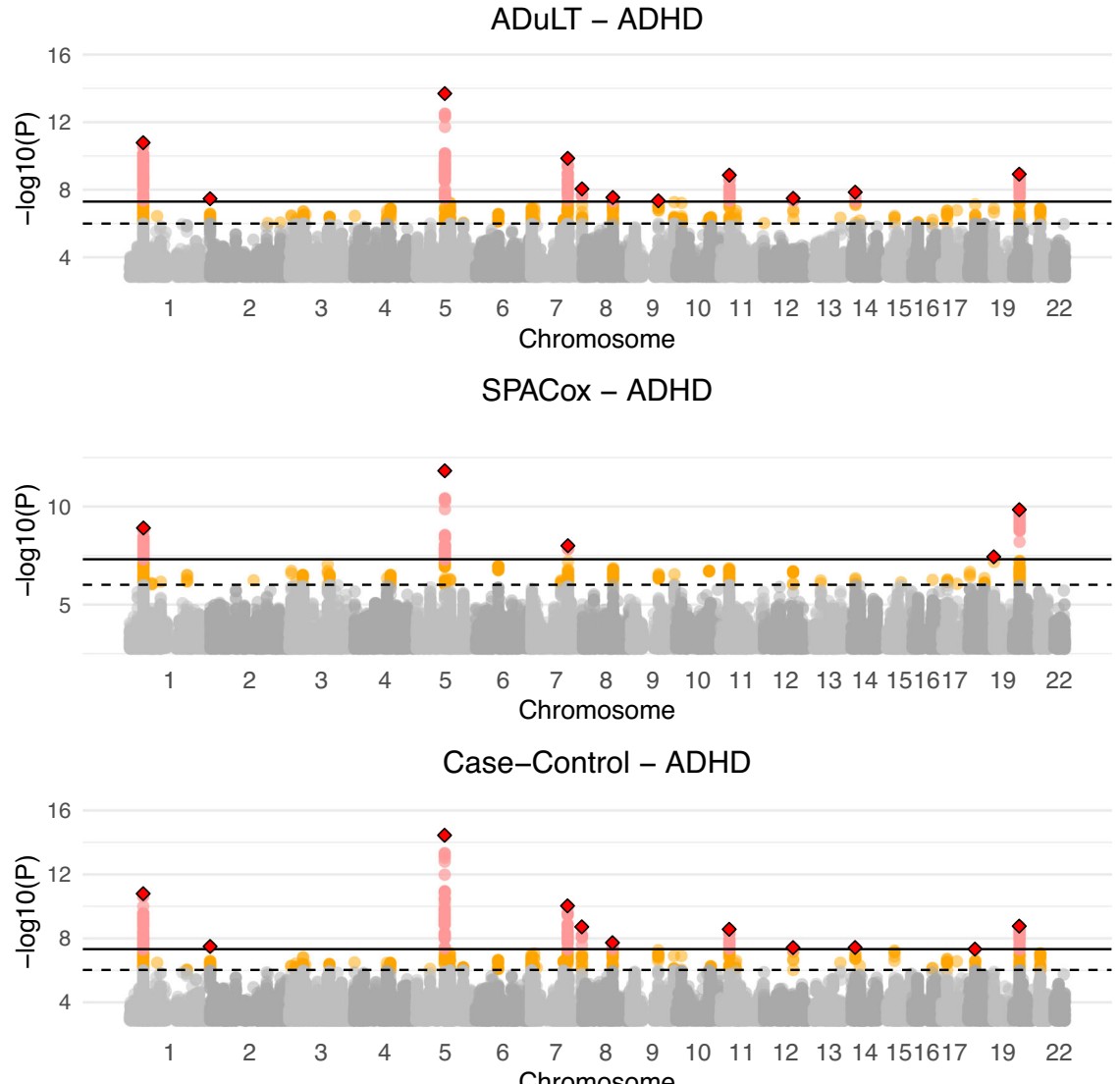

**Fig. 4 | Manhattan plots for ADHD for all three methods.** Case-control GWAS uses the age of individuals as a covariate, whereas the ADuLT GWAS and SPACox do not. The orange dots indicate suggestive SNPs with a *p*-value threshold of $5 \times 10^{-6}$. The red dots correspond to Bonferroni-adjusted genome-wide significant SNPs with a *p*-value threshold of $5 \times 10^{-8}$. The diamonds correspond to the lowest *p*-value LD clumped SNP in a 500k base pair window with an $r^2 = 0.1$ threshold. All tests performed are two-sided.

means ADuLT together with a linear regression may be faster and have higher power to detect causal SNPs.

## GWAS of psychiatric disorders in iPSYCH

The iPSYCH data has been linked to the Danish registers, which means that detailed information on age-of-onset, age, sex, and birth year can be assessed for all genotyped individuals that are part of the iPSYCH cohort[22]. This supplementary information was used to analyse four psychiatric traits, namely ADHD, autism, depression, and schizophrenia. For each of these phenotypes, population-based CIPs were obtained by birth year and sex (see Supplementary Figs. 18, 22, 26 and 30 for plots of the CIPs used, and see Cumulative Incidence Proportions for details). The prevalences were used to tailor the thresholds to each individual under the ADuLT model (see Methods).

We performed GWASs for each of the four phenotypes and for each of the methods considered, i.e. using either the case-control status or the estimated genetic liability by ADuLT as the outcome in a linear regression-based GWAS or SPACox (see Methods for details). Figure 4 displays the Manhattan plots for ADHD for all methods, where

the case-control GWAS included age as a covariate, while the ADuLT GWAS and SPACox did not. To report nearly independent findings, LD clumping was performed on the summary statistics with a $r^2$ threshold of 0.1 and a window size of 500kb, prioritising the SNPs with the lowest *p*-values. This was done for each combination of phenotype and method. The lowest *p*-value LD-clumped SNPs that are unique to ADuLT and ADHD can be found in Supplementary Table 3 and the LD-clumped snps that are unique to case-control status and ADHD can be found in Supplementary Table 4. For ADHD, we found 12 independent genome-wide significant associations when using the ADuLT phenotype as the outcome, while case-control status and SPACox found 11 and 5 associations, respectively. The ADuLT GWAS had two independent associations that were not identified by case-control associations, and case-control GWAS found one association that was not found by the ADuLT GWAS. One of the associations unique to ADuLT is rs4660756. The gene closest to this SNP is *ST3GAL3*, which has previously been associated with educational attainment[26] and ADHD[27]. SPACox also identified *ST3GAL3*, but through rs11810109 instead. The association unique to case-control GWAS is rs8085882 on

chromosome 18. The closest gene is *ZNF521*, which has previously been associated with education attainment[28], ADHD[29], and smoking initiation[30]. The association with the lowest *p*-value that is shared among all methods is rs4916723 on chromosome 5 with *LINC00461* as the closest gene. This gene has also been reported as being associated with educational attainment[30] and ADHD[31].

Across the four psychiatric disorders, ADuLT found 20 independent genome-wide significant associations, while case-control status found 17 and SPACox found 8. The Manhattan plots for each of the methods, each of the remaining disorders (autism, depression, and schizophrenia), and with and without age as a covariate can be found in Supplementary Figs. 19, 20, 23, 24, 27, 28, 31 and 32. We also provide the results of a GWAS with the ADuLT phenotype as the outcome and with both sex and age excluded from the model. The Manhattan plot can be found in Supplementary Figs. 21, 25, 29 and 33, and a summary table of all performed analysis can be found in Supplementary Table 2. Notably, SPACox consistently identified fewer associations than the ADuLT and case-control status GWASs, and was the only method that did not identify any significant association for major depression and schizophrenia. We also performed GWAS with the ADuLT phenotype where both sex and age were excluded from the phenotype. These analysis did not differ substantially from the presented analysis and can be found in Supplementary Figs. 20, 24, 28 and 32.

Finally, as SPACox does not provide effect sizes, we opted to plot *p*-values for validation since they are the only way to directly compare between GWAS models. We used the largest available summary statistics from PGC[1,32–34] for the four considered phenotypes, used the SNPs that were present in the iPSYCH data, and performed LD clumping (using the same settings as before) and prioritising the SNPs with the lowest *p*-values from the meta-analysed PGC summary statistics to keep only independent hits. Then, we took the SNPs that were genome-wide significant in the LD-clumped PGC summary statistics and plotted the *p*-values of the three considered methods against each other. The plot can be found in Supplementary Figure 34 and shows the *p*-values for ADuLT against case-control status and SPACox *p*-values on the SNPs that are independent genome-wide significant from the largest available PGC summary statistics, and they are therefore not necessarily genome-wide significant for either of the methods considered. The plot shows ADuLT has a slightly lower *p*-value than case-control and SPACox on the external genome-wide significant SNPs.

## Discussion

With biobanks such as the UK biobank[35], iPSYCH[22], FinnGen[36], or Biobank Japan[37] linking electronic health records to genetic data, there is an increased incentive to develop methods that can fully utilise this supplementary information. This includes details about age-of-onset, which can be used in time-to-event analyses to improve power. In epidemiology, time-to-event analyses are usually performed with a Cox-based regression, whereas time-to-event GWAS are still relatively uncommon. This has in part been due to computational challenges of applying Cox regression to GWAS, but recent developments of efficient Cox-based regression methods such as SPACox or GATE have largely resolved this limitation[9,11]. However, the performance of Cox-based regressions for GWAS has only been viewed in comparison to other Cox-based or logistic regression[5,8], and not when the case-control cohort is sampled with ascertainment (e.g. where cases are oversampled). Evaluating their performance in ascertained case-control cohorts is important as such datasets are very common in genetics, e.g. the iPSYCH and FinnGen data.

In this paper, we have examined the proportional hazards model implemented in SPACox and found that in situations where cases are ascertained (or oversampled; which is often the case in GWAS datasets), the proportional hazards based model was less powerful than a simple linear regression. We proposed the age-dependent liability threshold (ADuLT) model as an efficient and robust alternative to Cox-based time-to-event GWAS. The ADuLT model is the model underlying the recently published LT-FH++ method[20], as presented here it does not incorporate information on family members. However, the main focus of this paper was to compare the ADuLT model to a computationally efficient time-to-event GWAS method, SPACox, without accounting for information such as family history. ADuLT incorporates time-to-event information into the LTM by using liability thresholds that vary with age and sex. These personalised thresholds are derived from population-based estimates of the cumulative incidence proportions. Using this information, ADuLT first estimates individual posterior genetic liabilities, which are then used as a quantitative phenotype in GWAS. This final step can be performed with any continuous outcome GWAS software, which allows for ADuLT to benefit from using advanced GWAS methods, such as linear mixed models[13–15]. The computational cost of estimating the individual posterior liabilities is negligible when compared to the computational cost of performing even a simple GWAS with linear regression.

Using simulations, we compared different GWAS methods, Cox regression as implemented in SPACox and a linear regression with the ADuLT phenotype and the case-control status. As expected we found a Cox-based time-to-event GWAS to provide most power under the proportional hazards generative model, however it was closely followed by the ADuLT GWAS and case-control GWAS, especially when disease prevalence is low. Conversely, when simulating under the LTM, the ADuLT GWAS had the greatest power, followed by Cox regression and case-control GWAS. However, when considering ascertainment of cases, we found SPACox to have the lowest power of all considered methods under both generative models and for all prevalences except one (the least ascertained sample, i.e. the case ascertained data most similar to the full population). Interestingly, the simulations also show that Cox regression and ADuLT provide little or no benefit over standard case-control GWAS (linear or logistic regression) when the prevalence is small (e.g. <2%). We note that these results are in line with previously reported comparison between Cox regression and linear regression in case-cohort studies[8]. When we applied all three methods to the iPSYCH data, which has a high degree of case ascertainment, the results were in agreement with the simulation results in that SPACox identified fewer genome-wide significant variants than the case-control or the ADuLT GWASs. Therefore, for identifying significant genome-wide associated variants, a Cox-regression GWAS can have less statistical power than linear regression with case-control status or the ADuLT phenotype. As a result, we recommend using more robust GWAS methods, such as on case-control status or the ADuLT phenotype when performing GWAS in ascertained samples, which includes most case-cohort and case-control datasets.

Although Cox regression GWAS may not be robust to ascertained samples, we note that it can still improve power in population cohorts (i.e. population representative samples) for relatively prevalent outcomes. In addition, the ability to estimate unbiased individual absolute or relative risk over a time-period is also an important benefit of time-to-event models. Furthermore, several adjustments have been proposed to Cox regression when applied to ascertained data, such as inverse probability weighting[38] (IPW). IPW results in unbiased estimates, but estimating their variance (and association *p*-values) can be difficult[39]. Furthermore, to the best of our knowledge, IPW is currently not implemented in computationally efficient Cox regression GWAS methods (e.g. SPACox). Instead, we considered the proportional hazards implementation available in the `survival` package for R[40]. We used the proportional hazards model as generative model and ascertained the cases, but found no difference between the results from the survival package and SPACox, even with IPW. The results of the IPW can be seen in Supplementary Figs. 16 and 17. For time-to-event studies, where researchers are not able to design the study, one has to consider how participants have been sampled/recruited. Depending

on the sampling procedure, there are several potential biases present in the data and certain models may not be suitable. For instance, phenotypes in iPSYCH have been ascertained, and UKBB phenotypes are subject to collider bias and immortal time bias[41,42]. The appropriate model for either iPSYCH or UKBB is not a straightforward application of the proportional hazards model without properly considering and accounting for case ascertainment, collider bias, or immortal time bias. Instead a method tailored to each biobank and their particular sampling strategy as they each have different challenges. However, these biobank-specific challenges are often ignored in favour of a one-size-fits-all approach. With the sex and birth year stratified population-representative CIPs, ADuLT is able to account for these biases.

In contrast, we find ADuLT to be a computationally efficient and robust time-to-event GWAS method that in terms of statistical power performs on par with or better than Cox-regression GWAS in simulations. Moreover, using the LTM, it is possible to account for family history information[20,24], and it can be used in connection to risk prediction[43–45]. GWAS individual-level data can also be used to build polygenic scores based on efficient penalized regression models[46]; a future direction of research for us is to investigate whether a penalized linear regression using the ADuLT-inferred outcome would be preferable to using a Cox-based penalized regression as implemented in e.g. snpnet-Cox[47]. As other possible future directions, the ADuLT model may also provide an alternative framework for examining interactions between age and genetic variants[48], and provide insight into the genetics underlying disease trajectory. Like LT-FH[24] and LT-FH++[20], ADuLT also has the advantage that it produces quantitative posterior liabilities which can be treated as quantitative phenotypes and analysed with advanced GWAS method, such as BOLT-LMM[14], fastGWA[13], or REGENIE[15]. However, ADuLT does have some limitations. First, ADuLT requires population-representative CIPs to be available for the disorder of interest, and preferably stratified by sex and birth year. Recent efforts to make such data publicly available for all diseases is therefore of great interest[49]. If population-representative CIPs are not available, it is possible to use CIPs from a similar population. An example of this could be using population-representative CIPs from the danish registers in UKBB or FinnGen. However, in such a case, we would caution against fixing the full liability of a case, but rather only setting the lower limit to be the individualised threshold and letting the upper limit be infinite. Second, the assumption that early onset cases have higher disease liability may not always be true. Although age-of-onset tends to be negatively genetically correlated with case-control status, the correlation is not always very strong[50]. Third, the model does not account for possible interactions between genotype (or environment) with age, but exploring methods that model this relationship is a future direction. Fourth, similar to LT-FH[24] and LT-FH++[20], ADuLT assumes the narrow sense (additive) heritability is known a priori for the outcome of interest. These can either be obtained from literature or estimated in the data, e.g. using family-based heritability estimates[51]. However, we have also previously shown that the model we use is robust to misspecification of prevalence information and heritability[20]. Finally, in this study we did not consider downsampling of cases or ascertainment of healthy controls, which might be relevant for many genetic datasets such as the UK biobank[35] or the Danish blood donor study data[52].

It is not necessary to include age as a covariate in a GWAS with ADuLT as the outcome, since the effect of age is already accounted for in the phenotype itself. Traditionally, age or some related variable is included in the analysis to account for a person's lifespan and period of being at-risk. A common way to deal with such covariates in a regression is to project them out and consider only the univariate regression with the regressed outcome and predictor. Therefore, projecting out the covariates boils down to subtracting a value from the observations. This subtraction is not necessary with the ADuLT phenotype, as the effect of age and sex can be accounted for through the sex and birth year stratified CIPs. In fact, accounting for sex and age through the CIPs provide a more nuanced way of accounting for this information, as interactions are also considered. We have also performed the GWAS analyses without sex and age as covariates for the ADuLT phenotype, but it did not differ substantially from the other analysis.

Across the GWAS sample, the ADuLT phenotype is often bimodal in practice. The bimodality is due to the underlying truncated normal distributions leading to a gap between the resulting mean genetic estimates of the cases and controls. It has been shown that binary phenotypes when analysed with linear mixed models can suffer from inflation when the in-sample prevalence is low[15,53]. The ADuLT phenotype is quantitative, and is therefore not suitable for logistic regression (such as SAIGE or REGENIE), although it is often bimodal and may result in a clear separation between cases and controls. A potential solution is to employ rank-based inverse normal transformation to the ADuLT phenotype[54], but it may lead to a loss in power. Therefore, we recommend not using the ADuLT phenotype for GWAS when the in-sample prevalence is lower than roughly 1/80, which is in line with the advice provided by Mbatchou et al.[15] for the application of mixed linear models to binary outcomes.

As age information becomes more readily available, we expect time-to-event methods for GWAS that make use of such information to become more common. However, the benefit of these methods may depend on how the data was collected, as well as their ability to account for other confounders. We believe ADuLT provides both a robust, computationally efficient, and a flexible approach for time-to-event analyses of common diseases and outcomes in population-scale datasets.

## Methods

### The ADuLT model

The ADuLT model is an extension of the classical LTM[17,18], and is the model underlying our previously proposed LT-FH++ method[20]. To estimate an individual's genetic liability, ADuLT utilises birth year, sex, phenotype-specific age-of-onset for cases and current age for controls, as well as population-based cumulative incidences (i.e. the probability of having developed the disease at a given age). In contrast to LT-FH++, the ADuLT model does not incorporate family history as presented here. Instead, we focus on comparing ADuLT to standard time-to-event GWAS methods. ADuLT can account for cohort effects (changes in disease incidence by birth year), as well as differences by sex. This however requires population-based estimates to be available by age, sex, and birth year for each phenotype of interest.

The ADuLT model extends the classical LTM by allowing the threshold used to determine case-control status to depend on sex, birth year, and (if available) age-of-onset for an individual. The LTM assumes that each individual has a liability $\ell$ that follows a standard normal distribution in the population. When this liability is larger than a given threshold, $\ell \geq T$, where $\mathbb{P}(\ell \geq T) = K$ and $K$ is the trait's lifetime prevalence, then the individual is a case ($z = 1$), otherwise it is a control ($z = 0$). Under the ADuLT model, the trait prevalence $K$ is substituted by the population-representative CIP stratified by sex and birth year, if this information is available. If we let $s_i$ denote the sex of the $i$th individual and $b_i$ the birth year, then we can denote the sex and birth year stratified CIP as $K(t; s_i, b_i)$. It has the interpretation of being the proportion of individuals born in year $b_i$ and of sex $s_i$ that have been experienced the phenotype at time $t$. In Supplementary Fig. 26, an example of a CIP stratified by sex and birth year can be seen for depression. We can assign the personalised threshold in the following way:

$$\mathbb{P}(\ell_i > T_i) = K(t; s_i, b_i) \Rightarrow T_i = \Phi^{-1}(1 - K(t; s_i, b_i)), \quad (1)$$

where $\Phi$ denotes the CDF of the standard normal distribution. $T_i$ is then the $i$th individual's threshold. It is a function of sex, birth year, and

age or age-of-onset through the CIP, however this notation is suppressed for ease of notation. Additionally, we assume that the liability can be decomposed into two independent components, a genetic component, $\ell_g$, and an environmental component, $\ell_e$, such that $\ell = \ell_g + \ell_e$. The genetic liability $\ell_g$ is normally distributed with mean 0 and variance $h^2$, where $h^2$ denotes the trait heritability on the liability scale. The environmental component is normally distributed with mean 0 and variance $1 - h^2$ and independent of $\ell_g$.

ADuLT aims to estimate an expected genetic liability. We do this by expressing the liability as a 2-dimensional normal distribution given by:

$$\left(\ell_g, \ell\right)^T \sim N(\mathbf{0}, \boldsymbol{\Sigma}), \qquad \Sigma = \begin{pmatrix} h^2 & h^2 \\ h^2 & 1 \end{pmatrix}$$

The mean of the genetic component is given by

$$\mathbb{E}\left[\ell_g | z, h^2, K(t; s_i, b_i)\right]$$

where the information we condition on, namely the case-control status, heritability, and CIPs, result in an interval of (full) liabilities to integrate over. The CIPs fix the threshold at $T_i$ and excludes a range of possible liabilities, essentially resulting in a truncated multivariate normal distribution. The exclusion of liabilities can be thought of as "lived through risk". The case status determines if the integration is above or below the threshold. For a control, the interval $(-\infty, T_i]$ is considered. If an individual is a case and the full liability is fixed, then the interval of interest is simply $[T_i, T_i]$, i.e. a single value, and if the full liability is not fixed, we consider the interval $[T_i, \infty)$. The genetic liability estimated under the ADuLT model estimates the aggregated latent genetic liability and will be referred to as the ADuLT phenotype. Estimating the ADuLT phenotype does not require any genotype information, as it relies solely on phenotypic information. In Supplementary Fig. 4, the estimated genetic liability from ADuLT has been plotted against the true genetic liability. Performing a GWAS with the ADuLT phenotype is therefore a two-step procedure. First the ADuLT phenotype is estimated from phenotypic information, secondly a GWAS is performed with the ADuLT phenotype as the outcome. Any GWAS software that accepts continuous outcomes can be used with the ADuLT phenotype.

### Proportional hazards model

A proportional hazards model is commonly used to model the time to an event for various outcomes. It models the changes in the hazard function, which can be thought of as the instantaneous chance of experiencing the event at some point in time, $t$. The model commonly used for GWAS is given by

$$\lambda(t|X, G_j) = \lambda_0(t) \exp(\gamma X + \beta G_j) \qquad (2)$$

where $\lambda_0(t)$ is the baseline hazard, $X$ denotes the covariates, $\gamma$ is the covariate effects, $G_j$ is the genotype, and $\beta$ is the SNP effect. We note that a baseline hazard affects everyone, and the model can then examine the influence of covariates and the SNP in comparison to the baseline. The association test of interest is $H_0: \beta = 0$ vs $H_A: \beta \neq 0$. The baseline hazard is rarely known, but a common way to perform an association test in a proportional hazards model is with a likelihood ratio test, where the unknown baseline cancel out. A partial likelihood function is commonly used, which only maximises with respect to the variable of interest, here $\beta$.

However, maximum likelihood estimating can be very computationally expensive, which has been a limiting factor for most previous implementations of proportional hazards and mixed effects proportional hazards GWAS[10,55–57]. Recently, an efficient implementation of the proportional hazards model have been presented by Bi et al.[9], which allows for GWAS of large biobanks (>100.000 individuals). This

is achieved by use of the saddlepoint approximation. The null model (with $\beta = 0$) is determined only once, and the $p$-values are then efficiently calculated across the entire genome using the saddlepoint approximation on the null model's score test statistic. A trade-off of this approach is no effect sizes are available.

### Simulation details

In our simulations we use two generative time-to-event models, namely the Cox proportional hazards model and the age-dependent liability threshold (ADuLT) model. In the simulation results, we refer to the Cox proportional hazards as "Hazard", and the ADuLT as "Liability". We consider both of these generative models in order not to favour one time-to-event method over the other (i.e. SPACox and ADuLT GWAS). When referencing simulation results under both of these models, we will simply call them generative models, unless otherwise specified.

Initially, genotypes are simulated for $N = 1,000,000$ individuals and $M = 20,000$ independent SNPs. The genotypes are sampled from a binomial distribution $Binom(2, AF)$ with the probability parameter set to the allele frequency (AF) of a given SNP. The AFs are sampled from a uniform distribution on the interval $(0.01, 0.49)$. We chose not to include any low-frequency variants because the power to detect them in a GWAS setting is usually very small[58]. SNPs are standardised using the true AF, and for the scaled SNPs, the effect sizes of causal SNPs were drawn from the normal distribution $N(0, h^2/C)$, where $C$ denotes the number of causal SNPs and $h^2$ denotes the liability-scale heritability. Each SNP is simulated independently, which means we do not consider linkage disequilibrium in these simulations, in order to have a clear separation of causal and null SNPs. We used $h^2 = 0.5$ and either $C = 250$ or $C = 1000$ causal SNPs. For each choice of generative model and number of causal SNPs, we simulate 10 genotype data sets, which results in a total of 40 genotype data sets. With the simulated genotypes and causal effect sizes, we then generated synthetic phenotypes using the two generative models (details below). For each simulated genotype data set, phenotypes are derived under both generative models and for each disease prevalence considered. We consider four prevalences, three methods, and with and without case ascertainment.

Under the LTM, we set the trait status $z_i$ equal to 1 if the liability exceeds the threshold, i.e. if $\ell_i > T$, and 0 otherwise, where $\ell_i = X_i^T \beta + \epsilon_i = \ell_{g_i} + \epsilon_i$. The threshold $T$ is determined by the lifetime prevalence $K$. For instance, a lifetime prevalence of 5% and 10% results in thresholds $T = 1.64$ and $T = 1.28$, respectively. The relationship between the age-of-onset and the liabilities above the threshold $T$, is given by the logistic function

$$t_i(x) = \frac{K}{1 + \exp\left(-k(x - x_0)\right)}, \qquad (3)$$

where $K$ denotes the maximal attainable value, $k$ denotes the growth rate, and $x_0$ denotes the median age-of-onset. Using the age of controls, we know how long they have lived without being diagnosed. This information allows us to exclude liabilities, i.e. the period of risk lived through so far. For both cases and controls, the personalised thresholds are calculated as $T_i = \Phi(1 - CIP_i)$, where $T_i$ is the personalised threshold and $CIP_i$ is the CIP for individual $i$. The liabilities below the personalised threshold are considered for controls and the liabilities above the threshold are considered for cases. If the population-representative CIPs are stratified by birth year and sex, the full liability for cases can be fixed at $T_i$. Ages for controls are sampled from a uniform distribution between 10 and 90. This resulted in 90% of individuals having an age between 14 and 86.

For the proportional hazards model, we opted for a simulation setup as similar as possible to the one used in SPACox[9]. First, we simulated the censoring times, $c_i$, for each individual $i$ from an exponential distribution with a scale parameter of 0.15. Next, we simulated

onset times[59], $\tilde{t}_i$, using a Weibull distribution[60] as follows

$$\tilde{t}_i = \sqrt{\frac{-\log(U_i)}{\lambda \exp(\eta_i)}} \qquad (4)$$

where $\lambda$ is the event rate, $U_i \sim \text{Unif}(0,1)$, $\eta_i = X_i^T \beta + \epsilon_i$, with $\epsilon_i \sim N(0, 1 - h^2)$, and $X_i^T \beta$ are the scaled genotypes multiplied by effect sizes, corresponding to the genetic liability $\ell_g$ in the LTM. The case-control status $z_i$ is then 1 if $\tilde{t}_i < c_i$, and 0 otherwise. The event time $t_i = \min(\tilde{t}_i, c_i)$ is the observed time. The event rate $\lambda$ was chosen such that the lifetime prevalence is fixed at e.g. 1% or 5%. The simulation of onset times depends on all causal SNPs, which deviates from the simulations of onset times in the SPACox paper, where the onset times depended on a single causal SNP only. This change was made in order for the full genetic load of an individual to influence the onset times, instead of just a single SNP. Next, we calculated the CIP of the simulated event times, i.e. the fraction of cases observed before a given point in time, then the proportions were converted to the ages-of-onset (in years) using the logistic function given by Eq. (3) with median age-of-onset $x_0 = 50$ and growth rate $k = 0.2$. Both age and age-of-onset were used to calculate the cumulative incidence proportions, which in turn defines the thresholds under the ADuLT model. For instance, with a lifetime prevalence of 1%, 90% of all individuals had an age or age-of-onset between 17 and 57 years. The density plots of the simulated censoring and onset times can be found in Supplementary Figs. 1–3.

Case ascertainment is common in GWAS, as oversampling cases can increase statistical power. We therefore examine the impact of case ascertainment on each considered phenotype. In practice, we simulate case ascertainment by first simulating a full population of 1 million simulated genotypes and liabilities and case-control status (for different prevalences). We then apply case-ascertainment by drawing a sample of 20,000 individuals with a fixed case-control ratio of 50%, with 10,000 cases and 10,000 controls, regardless of what the population prevalence is in the full population.

## The ADuLT survival model

As we showed previously[20], the age-dependent liability threshold model can be considered a survival model. More specifically, consider the survival function $S_i(t) = \mathbb{P}(t_i > t)$, where $t_i$ represents age-of-onset for cases or censoring time for the $i^{th}$ individual, whichever happens first. Therefore, $t$ represents time on the scale of years. The probability that an individual has not become a case for a given $t$ is equal to the probability that the individual's liability is below the (individualised) liability threshold $T_i$, which is a shorthand notation for the age-dependent threshold given by Eq. (1).

If we assume that the individual liability consists of a genetic and an environmental component, $\ell_i = \ell_{g_i} + \ell_{e_i}$, where $\ell_{g_i}$ and $\ell_{e_i}$ are Gaussian distributed with mean 0 and variance $h^2$ and $1 - h^2$, respectively, then we can write the survival function as follows

$$S_i(t) = \mathbb{P}(t_i > t) = \mathbb{P}(\ell_i < T_i) = \Phi\left(\frac{T_i - \ell_{g_i}}{\sqrt{1 - h^2}}\right), \qquad (5)$$

where $\Phi$ is the standard Gaussian cumulative distribution function and we assume that the genetic liability contribution is known. In the last equality, we standardise the environmental contribution with the known genetic contribution and the variance. From this we can derive the event density, and the hazard function for the $i^{th}$ individual as

$$\lambda_i(age) = \frac{-S_i'(age)}{S_i(age)} \qquad (6)$$

We note that this survival model is unusual in a couple of ways. First, each individual has a slightly different parameterisation of the model, which comes through the individualised liability threshold $T_i$ from Eq.

(1). Second, the genetic effects affect the hazard rate by shifting the individual liability. Third, $T_i$ does not have to approach negative infinity as $t$ (the age-of-onset or age) approaches positive infinity, but may instead simply become fixed for all values $T_i$ above some threshold, e.g. if every individual in a cohort has died and no new event are possible. This is not necessarily a problem for the interpretation as $T_i$ may still be piece-wise differentiable, and the hazard rate for all values $t$ above this threshold then becomes 0.

## GWAS in iPSYCH

With the second wave of genotyped individuals, the iPSYCH case-cohort reached ~143,000 individuals, up from ~80,000[22]. Both waves have been imputed with the RICOPILI imputation pipeline[61], and were then combined into a single dataset. We restricted the analysis to SNPs that passed RICOPILI quality controls for both waves, resulting in a total of 8,785,478 SNPs for the GWAS. The analysis was restricted to a group of individuals with European ancestry, which were identified by calculating a robust Mahalanobis distance based on the first 20 PCs and restricting to a log-distance below 4.5[62]. We filtered for relatedness by removing individuals (the second one in each pair) with a KING-relatedness above 0.088. Since the iPSYCH case-cohort has a population representative subcohort and oversampled cases for six major psychiatric disorders (here we focus on ADHD, autism, depression and schizophrenia), we restricted each analysis to the individuals in the subcohort (which is a random sample of the entire population) and the cases for the phenotype being analysed, i.e. oversampled cases from the other psychiatric disorders were not used. The final number of individuals used for the GWAS of each phenotype is presented in Supplementary Table 1. The linear regression GWAS was performed using the bigsnpr package[25] for R and SPACox GWAS was performed using the original implementation in the SPACox package for R. We used 20 PCs, sex, and imputation wave as covariates for all analyses. We included age as a covariate when analysing case-control status. Age was not included as a covariate when using the ADuLT phenotype or SPACox. For SPACox, we did not include age as a covariate, as age was the timeline used. The ADuLT phenotype inherently accounts for age, which means it is not necessary to include as a covariate. We chose not to use a mixed model approach for GWAS with case-control status or ADuLT phenotypes, as SPACox did not have a similar option for random effects.

## Cumulative incidence proportions

The CIPs can be interpreted as the proportion of individuals diagnosed with a certain disorder before a given age. As a result, the CIPs are population and disorder specific and can be stratified by sex and birth year. The CIPs used here were stratified by sex and birth year to account for differences in incidences between sexes and for different birth years (cohorts). The CIPs were estimated from Danish population-based registers. The Danish Civil Registration System[63] was used to identify individuals and contains all 9,251,071 individuals that lived in Denmark at some point between April 2, 1968 and December 31, 2016. The Danish Civil Registration System has continually recorded information since its launch in 1968, and includes information about sex, date of birth, date of death, and date of emigration, or immigration. Each individual has a unique identifier that can be used to link information of several registers. Information on psychiatric disorders was obtained from the Danish Psychiatric Central Research Register[64]. It contains all admissions to psychiatric inpatient facilities since 1969 and visits to outpatient psychiatric departments and emergency departments since 1995. From 1969 to 1993, the International Classification of Diseases, eighth revision (ICD-8) was used as the diagnostic system. From 1994 onwards, the tenth revision (ICD-10) was used. The four disorders of interest were identified by the following ICD-8 and ICD-10 codes: ADHD (308.01 and F90.0), autism (299.00, 299.01, 299.02, 299.03 and F84.0, F84.1, F84.5, F84.8, F84.9), depression (296.09, 296.29, 298.09, 300.49 and F32, F33), and schizophrenia (295.x9 excluding 295.79 and F20). The age-of-

onset was defined as the age of an individual at first contact with the psychiatric care system, either inpatient, outpatient, or emergency visits. In the analyses, each individual was followed from birth, immigration, or January 1, 1969 (whichever happened last) until death, emigration, or December 31, 2016 (whichever happened first). The cumulative incidence function was estimated separately for each sex and birth year, and the Aalen-Johansen approach was used with death and emigration as competing events[65].

### Reporting summary
Further information on research design is available in the Nature Portfolio Reporting Summary linked to this article.

## Data availability
iPSYCH is approved by the Danish Scientific Ethics Committee, the Danish Health Data Authority, the Danish Data Protection Agency, Statistics Denmark, and the Danish Neonatal Screening Biobank Steering Committee[21]. Owing to the sensitive nature of the iPSYCH data, individual level data can only be accessed through secure servers where downloading individual level information is prohibited. International researchers may gain data access through collaboration with a Danish research institution. More information about getting access to the iPSYCH data can be obtained at https://ipsych.dk/en/about-ipsych. All datasets that can be shared are made available. The summary information from the simulations can be found in Supplementary Data 1 and at https://doi.org/10.6084/m9.figshare.22586239. The summary statistics for all performed GWAS in the project is available at the GWAS Catalog under the accession number GCP000675 at https://www.ebi.ac.uk/gwas/.

## Code availability
Code used to generate simulation results, analyse iPSYCH, and generate plots and tables can be found at https://github.com/EmilMiP/ADuLTCode. LT-FH++ can be found at https://github.com/EmilMiP/LTFHPlus. Both ADuLTCode and LTFHPlus are based heavily on packages such as dplyr and ggplot2 from the tidyverse packages found at https://www.tidyverse.org/. The code used to run SPACox is provided by its authors at https://github.com/WenjianBI/SPACox.

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

## Acknowledgements

We would like to thank Jakob Grove, Doug Speed, Matthew Robinson, and Sven Erik Ojavee for valuable discussions. Parts of Fig. 1 were created with BioRender.com. E.M.P, B.J.V. and F.P. were supported by the Lundbeck Foundation Initiative for Integrative Psychiatric Research, iPSYCH (R102-A9118, R155-2014-1724 and R248-2017-2003), and a Lundbeck Foundation Fellowship (R335-2019-2339). J.M., B.J.V. and F.P. were also supported the Danish National Research Foundation (Niels Bohr Professorship to Prof. John McGrath). A.J.S. is supported by a Lundbeckfonden Fellowship (R335-2019-2318), and O.P.-R. is supported by a Lundbeck Foundation Fellowship (R345-2020-1588). K.L.M. is supported by grants from The Lundbeck Foundation (R303-2018-3551) and the Brain & Behavior Research Foundation (Young Investigator Award 2021). A.G. has received funding from the European Research Council (ERC) under the European Union's Horizon 2020 research and innovation programme (grant agreement No 945733), starting grant AI-Prevent. High-performance computer capacity for handling and statistical analysis of iPSYCH data on the GenomeDK HPC facility was provided by the Center for Genomics and Personalised Medicine and the Centre for Integrative Sequencing, iSEQ, Aarhus University, Denmark (grant to A.D.B.). B.J.V. is also supported by Independent Research Fund (2034-00241B).

## Author contributions

E.M.P. performed the analysis. E.A., M.N., D.M.H., T.W., A.D.B., P.B.M., J.J.M., B.J.V, and F.P. performed sample and/or data provision and processing. E.M.P., F.P., and B.J.V. wrote the manuscript. E.M.P., E.A., O.P.-R., J.S., M.D.K., K.L.M., A.G., A.J.S., J.J.M., F.P., and B.J.V. performed core revision of the manuscript. F.P. and B.J.V. supervised the study. All authors contributed to critical revision of the manuscript.

## Competing interests

B.J.V. is on Allelica's international advisory board. The remaining authors declare no competing interests.

## Additional information

[1]National Centre for Register-Based Research, Aarhus University, Aarhus, Denmark. [2]Lundbeck Foundation Initiative for Integrative Psychiatric Research, iPSYCH, Aarhus, Denmark. [3]Centre for Integrated Register-based Research at Aarhus University, Aarhus, Denmark. [4]Department of Clinical Epidemiology, Aarhus University and Aarhus University Hospital, Aarhus, Denmark. [5]Institute of Biological Psychiatry, Mental Health Center - Sct Hans, Copenhagen University Hospital - Mental Health Services CPH, Copenhagen, Denmark. [6]Department for Congenital Disorders, Statens Serum Institut, Copenhagen, Denmark. [7]Department of Clinical Sciences, Copenhagen University, Copenhagen, Denmark. [8]Section for Geogenetics, GLOBE Institute, Faculty of Health and Medical Science, Copenhagen University, Copenhagen, Denmark. [9]CORE- Copenhagen Centre for Research in Mental Health, Mental Health Center—Copenhagen, Copenhagen University Hospital - Mental Health Services CPH, Copenhagen, Denmark. [10]Department of Biomedicine and iSEQ Centre, Aarhus University, Aarhus, Denmark. [11]Center for Genomics and Personalized Medicine, CGPM, Aarhus University, Aarhus, Denmark. [12]Department of Affective Disorders, Aarhus University Hospital-Psychiatry, Aarhus, Denmark. [13]Department of Clinical Medicine, Aarhus University, Aarhus, Denmark. [14]Institute for Molecular Medicine Finland, University of Helsinki, Helsinki, Finland. [15]Neurogenomics Division, The Translational Genomics Research Institute (TGEN), Phoenix, AZ, USA. [16]Queensland Brain Institute, University of Queensland, St Lucia, QLD, Australia. [17]Queensland Centre for Mental Health Research, The Park Centre for Mental Health, Wacol, QLD, Australia. [18]Bioinformatics Research Centre, Aarhus University, Aarhus, Denmark. [19]Novo Nordisk Foundation Center for Genomic Mechanisms of Disease, the Broad Institute of MIT and Harvard, Massachusetts, USA. [20]These authors contributed equally: Florian Privé, Bjarni J. Vilhjálmsson. ✉e-mail: emp@ncrr.au.dk; bjv@ncrr.au.dk

