## [Peer Review File · Nature Communications]

ADuLT: An efficient and robust time-to-event GWASREVIEWER COMMENTS

Reviewer #1 (Remarks to the Author):

Development of computationally efficient and robust statistical methods is crucial to advance genetic association studies in large-scale samples that can include hundreds of thousand individuals, for instance, such as UK Biobank. Comparison of such methods and selection of the most efficient and trustful one will shape further research activity and progress in this field.

In this manuscript, the authors tried to address two questions: (i) promote a modified version of their previously proposed approach, the age-dependent liability threshold model (ADuLT), and (ii) show that it is “a robust time-to-event GWAS method that performs on par with or better than” other methods when performing GWAS in large scale datasets. The authors performed analysis based on a few benchmarking samples, some of which were generated by the authors and others were drawn from the iPSYCH case-cohort sample. While the authors tried to address key problems for selecting future method(s) for large-scale GWAS, they did not provide clear description neither of the methods, phenotypes and samples used in the underlying analyses, nor of the obtained results, which leads to the lack of transparency of this study and prevents reproducibility of the claimed results. Unfortunately, the presentation approach used by the authors in this manuscript can lead the readers to false or biased conclusions.

I would like to mention about several issues of the present study.

(1) The authors stated that they analyzed “ADuLT phenotype”, but they did not provide any description of this phenotype in the manuscript.

(2) The authors presented comparison of the results obtained by different methods in different samples (see Figure 2), but did not provide clear descriptions about how these samples were drawn (i.e. which criteria or selection rules were used), how many simulations were performed and for which number of samples. Moreover, the authors did not provide any details about what is displayed in Figure 2.

(3) One of the main conclusions of the manuscript (see Figure 3), i.e. that the ADuLT method “performs on par with or better than” other methods when performing GWAS in large scale datasets, was based on comparing parallelized version of this method with not-parallelized versions of other methods. Moreover, the figure caption does not correspond to the items displayed in Figure 3.

I would suggest to revisit the manuscript substantially by making it clearer and more transparent.

I hope that addressing other issues listed below can help to improve the manuscript considerably too.

Other comments

(4) In this manuscript, the authors demonstrated that the ADuLT method, which is promoted by the authors to be used in GWAS analyses of large-scale studies, did not provide any advantages as compared to commonly adopted case-control approaches.

Why did the authors not state about this finding in the abstract and in conclusion section?

(5) In Ref. [5] it was demonstrated that the SPACox approach perfectly control for type-I errors. Larger number of GWAS significant SNPs that the ADuLT approach detected in this study can be related to type-I error.

Could the authors clearly demonstrate that the ADuLT method does not introduce additional type-I error, which can lead to the larger number of the detected GWAS significant SNPs and inflation of GWAS summary statistics?

(6) The authors reported that the ADULT approach identified a larger number of GWAS significant SNPs. It could be related to the winner effect. It would be helpful for the reader if the authors could provide further estimate(s) and insight about the winner effect in their study.

(7) The following conclusion made by the authors has a restricted power and applicability:

"Therefore, for identifying significant genome-wide associated variants, a Cox-regression GWAS can have less statistical power than linear regression with case-control status or the ADuLT phenotype."

On the other hand, the authors mentioned advantages of the Cox-regression approaches:

"Although Cox regression GWAS may not be robust to ascertained samples, we note that it can still improve power in population cohorts and may still yield unbiased estimates."

I would propose to clearly state areas where ADuLT, case-control and Cox-based approaches are preferred methods for scientific investigations.

(8) The study is not well designed [thought out]. For instance, could the authors describe how case-control samples can be differently drawn from populations with different prevalence (see Figure 2)? Respective inclusion/exclusion criteria can be a source of bias of the present study.

(9) In section 2.1, the authors represented "Model", but not models. Why did the authors not present other approaches as well, if the aim was to compare ALL these approaches but not providing preference for one of them, the ADuLT?

(10) The authors made contradictory statements:

(A) In the abstract they mentioned that ADuLT is "a robust time-to-event GWAS method":

"We find that ADuLT to be a robust time-to-event GWAS method that performs on par with or better than Cox-regression GWAS"

(B) In the Methods section the authors wrote that the ADuLT model is an extension of the classic LTM, which "does not account for time-to-event." (see second paragraph in section 2.1)

Could the authors clarify and, possibly, rewrite respective parts of the manuscript by clearly describing whether the ADuLT model is a time-to-event model, or it is something else, and briefly but clearly demonstrate/describe how this model takes into account time-to-event information?

(11) Second paragraph in section 2.2. The authors wrote:

"The AFs are sampled from a uniform distribution on the interval (0.01, 0.49)."

This means that the authors overrepresented genetic markers with high MAF, because it is known that the number of genetic markers with low MAF is substantially higher than that with large MAFs and their simulations do not represent real distribution of MAFs within human genome.

(12) The authors simulated two distributions, namely, for censoring time c_i and for event time t^{\sim}_{i} , but they did not provide any figure demonstrating how these two distributions are related to each other.

It makes difficult for the readers (considering a broad spectrum of interests of the readers of the journal) to imagine what was simulated by the authors.

It would substantially improve the manuscript, if the authors could introduce figures with respective distributions, helping the readers to visualize the proposed approach and simulations used when making the conclusions.

(13) Definition of liability threshold in section 2.3 seems contradictory.

From the first three paragraphs, it follows that ALL individuals who have the same birth year, sex and age should have the same liability threshold. The latter means that ALL these individuals are either cases or controls, which is not true and only a part of them are cases and the other part are controls, otherwise the prevalence is equal either to 0 or 100%.

(14) The authors used terms "the generative models" but did not distinguish and clearly state when they generated data by using respective models or when they analyzed real data.

(15) The authors did not provide any information about how many repeated simulations were performed to obtain results presented in Figure 2 and Figures S1-S6.

(16) In Figures S4-S7, the authors presented result for "Power ratio", but they did not explain what it means. Therefore, these Figures do not provide any additional information.

Moreover, it is difficult to understand the titles of these Figures, in particular their meaning.

(17) Figure 3 is misleading. The authors refer to "bars" in the caption, but there are no any bar displayed in this figure. Moreover, abbreviation "M" was not defined.

(18) In section 3.2, the authors refer to "the average χ^2 -statistics", but they did not provide any explanation about its meaning, and why in Figures S2-S3 and S6-S7 they refer to "average null χ^2 ". What is the difference?

(19) In the same section the authors use term "the null SNPs". They did not provide any information about its meaning. It is a jargon, which were used in a few publications, but which was not accepted by broader scientific community.

(20) In a sentence before subsection 3.2.1, the authors wrote:

"In Table S1, simulation results for all parameter setups are available, including the power, relative power compared to SPACox, and mean chi-square statistic of null SNPs."

Could the authors specify where the reader can find "relative power" and "mean chi-square statistic of null SNPs" in Table S1?

Reviewer #2 (Remarks to the Author):

This paper introduces ADuLT, a new version of a liability model for GWAS data, and compares its performance with Cox regression and logistic regression, focusing particularly on the setting of case-control or case-cohort data where cases are over-represented relative to the population compared to controls. I found the description of the new method somewhat confusing, in part because it seems to be described partly in the methods section and again at the start of the results section. I think I get the basic idea, however: namely, that each case is assigned a “liability threshold” based on the cumulative normal distribution for their age, sex, year of birth (and potentially other variables) using population cumulative incidence rates, and this value is used as a continuous phenotype (for cases, or one minus that for controls); this is then used as the outcome variable in a standard regression model with the genotypes as covariates. Although in the simulation section, it is clear that that entire vector of genotypes, multiplied by their respective effect sizes, is used as the predictor, it isn’t clear in the analysis whether each gene is analyzed separately or some multivariate model (which obviously would require some kind of penalization).

My bigger concern has to do with the implementation of the Cox model. For unsampled data, this would be straightforward, although computationally intensive if the full maximum likelihood or Wald test is used, but less so if the score test (which doesn’t require iteration) is used; each gene could be analyzed separately after adjusting for a common set of covariates. (worth explaining that as age is the time scale, including it as a covariate would be unnecessary!). The same computational efficiency by using score tests would apply to logistic regression. What confuses me is the application to sampled case-control data. The appropriate analysis for nested case-control data would then be conditional logistic regression, comparing each case to its sampled risk set members; for case-cohort data, it would be a modified Cox regression comparing each case to the surviving subcohort members, not including the other cases that weren’t part of the subcohort, with an appropriate adjustment to the variance estimator. For unmatched case-control data, the appropriate analysis would be unconditional logistic regression, including age as a covariate, but Cox regression without any allowance for the sampling would not be appropriate anyway, hence its poor performance in the sampled simulation settings. So a clearer description of how Cox regression is used is needed.

Minor points

Reference [1] is missing authors. Reference [11] is missing the journal.

P 4 sec 2.2: explain that each genotype is sampled independently, without considering LD. Also in the middle of the second paragraph, the sentence “This model does not account for time-to-event.” Is misleading, as age is part of the individualized liability threshold (via CIP(age)). Ditto lines 3-4 from the end of sec. 2.4.

P 5 line 7 up: Eq.[1] hasn't appeared yet.

Middle of p. 13: change neglectable to negligible. Further down, "the least ascertained sample" seems misleading, as I think what you mean is the highest prevalence sample, so if all cases are included, largest proportion of controls are sampled.

P 14: the sentence "the assumption that early onset cases have higher disease liability may not always be true", seems to conflate two separate concepts. Obviously as the CIP increases with age, the marginal liability threshold decreases. In a proportional hazards model, the genetic HR is assumed to be constant, but for some diseases (e.g., breast cancer), the genetic HR is larger at younger ages.

Reviewer #3 (Remarks to the Author):

General

The authors presented ADuLT, an age-dependent liability threshold model for GWAS. Through both simulation and analyses of data from iPSYCH the authors have demonstrated that ADuLT is more powerful, robust and faster to run than Cox regression models, especially when cases are strongly ascertained (as in iPSYCH). This paper presents an interesting and useful addition to the current GWAS methods, which I think will be very useful for the field going forward.

Major

1. As the authors described in the introduction, Methods and Results, ADuLT calculates a personalised liability threshold for case-control status inference for all individuals using population-based cumulative incidence proportions (CIPs) for the phenotype of interest as a function of age, sex and birth year. For iPSYCH, the authors derived CIPs from the Danish population registers. CIPs seem only possible to calculate if one has population based data on disease incidence by age; the authors mentioned in various points in the paper that this is becoming more widely available in datasets linked to electronic health records and registers (i.e. independent on the GWAS data). I wonder if the authors thought about generating these CIPs using GWAS data alone (i.e. age of individuals in the GWAS), or using non-population matched CIPs (i.e. using Danish CIPs on UKBiobank for example). It would be ideal to see one analysis to show what would happen in each case, to demonstrate the problems and considerations for each of these non-standard applications of ADuLT. In addition, a brief discussion would be very helpful for users who want to apply this model but do not have population matched CIP.

2. The authors performed GWAS on ADHD, SCZ, ASD and MDD in iPSYCH using ADuLT, and compared its results to GWAS performed using SPACox and Case-control logistic regression, concluding that ADuLT gives more hits (20 independent GWAS hits rather than 17 in case-control and 8 in SPACox). The authors also gave some instances of ADHD hits found only in ADuLT that has been previously reported for ADHD. I would like to see forest plots of the 20 hits found across the 4 diseases for their effects in external datasets (like PGC, UKBiobank, FinnGen etc) to know that across the board the hits obtained by ADuLT are credible. It would be great to compare the replication rate of ADuLT hits vs those found in SPACox and case-control.

3. Why was sex added as covariates in ADuLT GWAS (and age not), when it is (both are) already used to define the liability threshold of each individual? Why should/shouldn't it matter? Please explain. The supplementary materials showed manhattans with and without adding age as covariate, so clearly the authors tried to find out if adding in age as covariate made a difference - did they try this just to be consistent with covariates selected for case-control GWAS? Turns out adding in age or not for ADuLT seems to matter a tiny bit (few hits different), which is not obvious unless I looked really carefully - suggest presenting in a table summarising results with/without age as covariate in addition to manhattans.

4. Simulations and its associated figures (figure 2, bunch of supplementary figures): the down-sampling experiment was to examine the effect of ascertainment of cases - why not just call it "ascertained", to be crystal clear? I had to tell myself downsampling really means ascertainment when I read figure 2 and all similar supplementary figures, as downsampling alone (if non-ascertained) is not going to result in power differences we see in those figures, ascertainment would.

REVIEWER COMMENTS

This colour is used for reviewer comments.

This colour is used for replies to reviewer comments.

Changes to the text have been marked in red.

Reviewer #1 (Remarks to the Author):

Development of computationally efficient and robust statistical methods is crucial to advance genetic association studies in large-scale samples that can include hundreds of thousand individuals, for instance, such as UK Biobank. Comparison of such methods and selection of the most efficient and trustful one will shape further research activity and progress in this field.

In this manuscript, the authors tried to address two questions: (i) promote a modified version of their previously proposed approach, the age-dependent liability threshold model (ADuLT), and (ii) show that it is “a robust time-to-event GWAS method that performs on par with or better than” other methods when performing GWAS in large scale datasets. The authors performed analysis based on a few benchmarking samples, some of which were generated by the authors and others were drawn from the iPSYCH case-cohort sample. While the authors tried to address key problems for selecting future method(s) for large-scale GWAS, they did not provide clear description neither of the methods, phenotypes and samples used in the underlying analyses, nor of the obtained results, which leads to the lack of transparency of this study and prevents reproducibility of the claimed results. Unfortunately, the presentation approach used by the authors in this manuscript can lead the readers to false or biased conclusions.

We would like to thank the reviewer for framing and highlighting the importance of our work. We are sorry if we have not provided enough details in the description of our study, or if some results have been presented in a confusing manner. We have tried our best to improve on this in the revision.

I would like to mention about several issues of the present study.

(1) The authors stated that they analyzed “ADuLT phenotype”, but they did not provide any description of this phenotype in the manuscript.

The “ADuLT phenotype” referenced is the estimated genetic liability under the ADuLT model. We have now added the following sentence to the Methods section to clarify.

The estimated genetic liability under the ADuLT model estimates the aggregated latent genetic liability and will be referred to as the ADuLT phenotype.

(2) The authors presented comparison of the results obtained by different methods in different samples (see Figure 2), but did not provide clear descriptions about how these samples were drawn (i.e. which criteria or selection rules were used), how many simulations

were performed and for which number of samples. Moreover, the authors did not provide any details about what is displayed in Figure 2.

We apologise for this confusion and have now added more details in the simulation details section to improve clarity of the manuscript.

- Clarified the number of simulations done
- Clarified the number of genotype data sets simulated
- What we mean by liability and hazard in the plots
- What we mean by generative model
- Added the censoring and onset times to the supplemental notes and added a reference to these in the simulation details. The plots include:
 - All onset times and censoring times for all prevalences
 - Observed onset and censoring times for all prevalences
 - The event times (ignoring case status) for all prevalences
- Clarified how we obtain case ascertainment in the simulations

The added description of how we achieve case ascertainment through downsampling is described below:

Case ascertainment is common in GWAS, as oversampling cases can increase statistical power. We therefore examine the impact of case ascertainment on each considered phenotype. In practice, we simulate case ascertainment by first simulating a full population of 1 million simulated genotypes and liabilities and case-control status (for different prevalences). We then apply case-ascertainment by drawing a sample of 20,000 individuals with a fixed case-control ratio of 50%, with 10,000 cases and 10,000 controls, regardless of what the population prevalence is in the full population.

(3) One of the main conclusions of the manuscript (see Figure 3), i.e. that the ADuLT method “performs on par with or better than” other methods when performing GWAS in large scale datasets, was based on comparing parallelized version of this method with not-parallelized versions of other methods. Moreover, the figure caption does not correspond to the items displayed in Figure 3.

We apologise for the confusion. The wording “performs on par with or better than” Cox-regression GWAS in the abstract refers to power comparisons in the simulations and previously identified associations in published GWASs, and not to the runtime results. We have now modified the sentence to clarify this.

We find ADuLT to be a computationally efficient and robust time-to-event GWAS method that, in terms of statistical power, performs on par with or better than Cox-regression GWAS in simulations.

Regarding the computational efficiency of ADuLT, the reviewer is correct that we compared it with a parallelized version of ADuLT. We now provide the comparison using different numbers of cores in an updated version of **Figure 3**. We however note that, as far as we can see, the current implementation of SPACox cannot use more than one core. We therefore deliberately want to highlight that ADuLT is highly parallelizable, as it is in our opinion a major strength of ADuLT.

Regarding the caption for Figure 3, we apologise for the confusion and have now updated it and its caption in the hope to improve clarity.

Figure 3: Running times of ADuLT combined with a linear regression GWAS compared to SPACox.

Each point represents the mean value of 10 replications, while the error bars are represented by the estimate ± 1.96 standard errors (of the mean). Run times were assessed for a varying number of individuals and SNPs. The number of SNPs (M) varies from 100k to 1 million, while the number of individuals vary from 10k to 100k. SPACox uses a single CPU core, as no parallelization is available. We used 1, 2, and 4 CPU cores for estimating the ADuLT phenotype and performing the linear regression GWAS for this phenotype. The means and corresponding standard errors of the run times can be found in Table S1.

I would suggest to revisit the manuscript substantially by making it clearer and more transparent.

We believe that the many changes suggested by the three reviewers have substantially improved the clarity and transparency of the manuscript.

I hope that addressing other issues listed below can help to improve the manuscript considerably too.

Other comments

(4) In this manuscript, the authors demonstrated that the ADuLT method, which is promoted by the authors to be used in GWAS analyses of large-scale studies, did not provide any advantages as compared to commonly adopted case-control approaches. Why did the authors not state about this finding in the abstract and in conclusion section?

Yes, we agree that this important result should have been highlighted and discussed in the conclusion. We have now modified the abstract, the introduction, and the conclusion section to highlight that the benefit of time-to-event GWAS methods for outcomes with low prevalence (<2%) is expected to be small (and provide the following citation for that [DOI: 10.1016/0021-9681(83)90165-0]). In this case a standard case-control GWAS (e.g. logistic regression) already provides sufficient power.

The abstract now says

As more genetic data are being linked to electronic health records, robust GWAS methods that can make use of age-of-onset information have the opportunity to increase power in analyses for common health outcomes.

The introduction now says

The frailty model inherits some of its advantages from the mixed model[53, 22, 30, 32], and can both account for population structure and relatedness, as well as improve statistical power when sample sizes are large. A third reason is that time-to-event models are generally not expected to provide significant gains in power for rare health outcomes [DOI: 10.1016/0021-9681(83)90165-0]. Indeed, the performance of Cox-based regressions in a GWAS setting is poorly understood, and they have only been viewed in comparison to other Cox-based regressions or logistic regression[48, 19].

The conclusion now says

However, when considering ascertainment of cases, we found SPACox to have the lowest power of all considered methods under both generative models and for all prevalences except one (the least ascertained sample). Interestingly, the simulations also show clearly that Cox regression and ADuLT provide little or no benefit over standard case-control GWAS (linear or logistic regression) when the prevalence is small (e.g. <2%). We note that these results are in line with previously reported comparison between Cox regression and linear regression in case-cohort studies[48].

also

We believe ADuLT provides both a robust, computationally efficient, and a flexible approach for time-to-event analyses of common diseases and outcomes in population-scale datasets.

(5) In Ref. [5] it was demonstrated that the SPACox approach perfectly control for type-I errors. Larger number of GWAS significant SNPs that the ADuLT approach detected in this study can be related to type-I error.

Could the authors clearly demonstrate that the ADuLT method does not introduce additional type-I error, which can lead to the larger number of the detected GWAS significant SNPs and inflation of GWAS summary statistics?

We have now elaborated on the simulation studies by calculating the false positive rate for different levels of significance thresholds. Before, we only had the false positive rate at the genome-wide significance level of 5×10^{-8} . Now it is available for 0.05, 0.005, ..., and 5×10^{-7} as well as 5×10^{-8} . A table with these values has been added to **Table S1** in a new sheet called "False Positive - Simulations". The caption of the table has therefore been changed to:

new table title:

Excel file with summary information from the simulations, containing information such as power, average of the chisq-statistics of the null SNPs, false positive rates for varying significance levels, etc

new table caption:

All summary information from the simulations has been combined into this Excel file and it contains several sheets. We explain what information each sheet holds and what the important columns are called. The first sheet is the raw simulation results for each replication (v). This includes information on power (power), number of false positives (FP) and causal SNPs (causal) identified at 5×10^{-8} . The average χ^2 -statistics are also provided for the null SNPs (mean_null_chisq) and causal SNPs (mean_causal_chisq). This information is given for each choice of generative model (gen_mod), prevalence in the full data (prev), model used to assign phenotype under the generative model(method), total number of causal SNPs (C), and whether case ascertainment was present or not (downsampling). Each simulation setup has 10 replications, and information is available for each iteration. The second sheet provide averages across replications for a set of parameters. The average power (avg_power) and null χ^2 -statistic (avg_null), as well as their standard error (avg_power_se and (avg_null_se) and number of replications that did not identify any of the causal SNPs (no_causal) is reported. The third sheet provide the run-times plotted in Figure 3. The number of individuals (N), SNPs (M), and cores used (ncores) are reported. The average run times (mean_times) and the standard error (se_times) are reported. The fourth sheet provide false positive rates (FP_prop) and standard errors (FP_sem) for a significance thresholds ranging (alpha_lv) from 0.05, 0.005, to 5×10^{-8} . The number of false positives at a given significance threshold is given by (FP).

and the simulation results section have been updated to reflect this as well, and it now reads:

All methods controlled well for type-1 errors (false positives) at varying significance levels. No false positives were observed for any method at a significance level of 5×10^{-8} . All false positive results can be found in Table S1.

We do not see any inflation in the false positive rate for ADuLT and the average chi-square statistic is well-calibrated at 1. Therefore, we do not believe that the higher number of genome-wide significant SNPs in the iPSYCH data is due to inflation or not being properly calibrated, but is due to power improvement.

(6) The authors reported that the ADuLT approach identified a larger number of GWAS significant SNPs. It could be related to the winner effect. It would be helpful for the reader if the authors could provide further estimate(s) and insight about the winner effect in their study.

Although we use a 2-stage approach, it should not lead to a winner's curse as we only use age, age-of-onset, sex, and birth-year to obtain the ADuLT phenotypes and not individual-level genetic data. Therefore, we do not expect estimated effect sizes to be biased on average. We now note the two-step nature of the method explicitly in the methods section.

(7) The following conclusion made by the authors has a restricted power and applicability: "Therefore, for identifying significant genome-wide associated variants, a Cox-regression GWAS can have less statistical power than linear regression with case-control status or the ADuLT phenotype."

On the other hand, the authors mentioned advantages of the Cox-regression approaches: "Although Cox regression GWAS may not be robust to ascertained samples, we note that it can still improve power in population cohorts and may still yield unbiased estimates."

I would propose to clearly state areas where ADuLT, case-control and Cox-based approaches are preferred methods for scientific investigations.

Yes, we see how this can seem contradictory. We have now modified the conclusion to clarify this better. Together with the changes addressing comment (4) we believe it should now be clearer when ADuLT is useful.

Although Cox regression GWAS may not be robust to ascertained samples, we note that it can still improve power in population cohorts (i.e. population representative samples) for relatively prevalent outcomes. The ability to estimate unbiased individual absolute or relative risk over a time-period is also an important benefit of time-to-event models.

(8) The study is not well designed [thought out]. For instance, could the authors describe how case-control samples can be differently drawn from populations with different prevalence (see Figure 2)? Respective inclusion/exclusion criteria can be a source of bias of the present study.

In our reply to Reviewer #3, question 1, we discuss the cumulative incidence proportions (CIPs) for a none-population matched analysis, such as using CIPs from the Danish registers in UKBB or FinnGen. The sex and birth year stratified CIPs we use are able to account for potential biases from selection, as they are population representative. The in-sample analysis is therefore not biased by selection criterias.

(9) In section 2.1, the authors represented "Model", but not models. Why did the authors not present other approaches as well, if the aim was to compare ALL these approaches but not providing preference for one of them, the ADuLT?

We have added a subsection (now section 2.2) on the Cox PH model to the Methods section. We also added a paragraph on why SPACox was used, namely the computational efficiency.

(10) The authors made contradictory statements:

(A) In the abstract they mentioned that ADuLT is "a robust time-to-event GWAS method":

"We find that ADuLT to be a robust time-to-event GWAS method that performs on par with or better than Cox-regression GWAS"

(B) In the Methods section the authors wrote that the ADuLT model is an extension of the classic LTM, which "does not account for time-to-event." (see second paragraph in section 2.1)

Could the authors clarify and, possibly, rewrite respective parts of the manuscript by clearly describing whether the ADuLT model is a time-to-even model, or it is something else, and briefly but clearly demonstrate/describe how this model takes into account time-to-even information?

We do agree with the reviewer that the liability threshold model (LTM) does not account for time-to-event, as it assumes a fixed thresholded. However, ADuLT provides an extension of the classic liability threshold model, such that every individual has a unique threshold based on their birth year, sex, and age or age-of-onset. This LTM extension (ADuLT) makes it a time-to-event model as well. To avoid misunderstandings, we have modified the text to only say if a model is a time-to-event model rather than say when it is not.

(11) Second paragraph in section 2.2. The authors wrote:

"The AFs are sampled from a uniform distribution on the interval (0.01, 0.49)."

This means that the authors overrepresented genetic markers with high MAF, because it is known that the number of genetic markers with low MAF is substantially higher than that with large MAFs and their simulations do not represent real distribution of MAFs within human genome.

Yes, the simulated distribution does not match typical MAF distributions in GWAS data. Below you can see the plots for 250 causal SNPs and with case ascertainment. We have now examined the effect of the MAF and effect sizes on the power. We chose not to include any low-frequency variants (<1%) because the power to detect them in a GWAS setting is usually small (see e.g. <https://doi.org/10.1038/nrg3706>). .

Power to detect causal variants at a threshold of 5×10^{-8} , as a function of effect size. The power curve is calculated by arranging the absolute value of the effect sizes by size and calculating the proportion of true positive associations. The results are pooled across 10 replications.

Power as a function of MAF. The power curve is calculated by arranging the MAFs by size and calculating the proportion of true positive associations. The results are pooled across 10 replications.

(12) The authors simulated two distributions, namely, for censoring time c_i and for even time t^{\sim}_{i} , but they did not provide any figure demonstrating how these two distributions are related to each other.

It makes difficult for the readers (considering a broad spectrum of interests of the readers of the journal) to imaging what was simulated by the authors.

It would substantially improve the manuscript, if the authors could introduce figures with respective distributions, helping the readers to visualize the proposed approach and simulations used when making the conclusions.

We agree that this could be explained better, and that providing a figure is a good idea. We have now plotted the simulated censoring times and event times and put them in the supplementary materials. The plot is shown below (now added to the manuscript as a supplementary figure), and the simulation details regarding the Cox PH model now has a reference to them.

Simulated censoring and age-of-onset times for varying prevalences

For all 1 million simulated individuals, a censoring, c , and onset \tilde{t} , is simulated. For a sub-sample of 20,000, we illustrate both here for different prevalences. Here, **Censor** refers to the censoring time in all individuals and **Onset** refers to the onset times. An individual is only a case if the onset time occurs before the censoring time.

A similar plot of the observed censoring time and observed onset time is also available in the supplementary material, both colored by case status (as the one shown above) and as a simple histogram of the observed event times (both censoring and onset times combined).

(13) Definition of liability threshold in section 2.3 seems contradictory.

From the first three paragraphs, it follows that ALL individuals who have the same birth year, sex and age should have the same liability threshold. The latter means that ALL these

individuals are either cases or controls, which is not true and only a part of them are cases and the other part are controls, otherwise the prevalence is equal either to 0 or 100%.

It is true that individuals that have the same birth year, sex, and age or age-of-onset will have the same liability threshold. However, the individual liability varies and only the fraction of individuals with liabilities larger than their threshold become cases. We have added a sentence in the overview of methods to better explain this. It now says

The ADuLT model modifies the LTM by assuming that the threshold used to determine an individual's case-control status corresponds to the CIP at the age of diagnosis. Only individuals with liabilities above their assigned liability threshold, which depends on their age, sex, and birth year, become a case.

(14) The authors used terms "the generative models" but did not distinguish and clearly state when they generated data by using respective models or when they analyzed real data.

We apologise for this confusion. The two generative models that we refer to are the ADuLT model and the proportional hazards (PH) model. These models are listed in the simulation benchmark figures (e.g. Figure 2) as "Liability" (i.e. ADuLT) and "Hazard". To make this notation more clear, we have now extended the simulation description.

In our simulations we use two generative time-to-event models, namely the Cox proportional hazards model and the age-dependent liability threshold (ADuLT) model. In the simulation results, we refer to the Cox proportional hazards model as "Hazard", and the ADuLT model as "Liability". We consider both of these generative models in order not to favour one time-to-event method over the other (i.e. SPACox and ADuLT GWAS). When referencing simulation results under both of these models, we will simply call them generative models, unless otherwise specified.

(15) The authors did not provide any information about how many repeated simulations were performed to obtain results presented in Figure 2 and Figures S1-S6.

We would like to thank the reviewer for pointing this out. We are not sure how we managed to miss this detail during the internal review process. We used 10 replications for each parameter setup, and the information has been added to both the simulation details and simulation results sections. We added the following to the second paragraph (the one describing the simulated genotypes) of the simulation details:

For each choice of generative model and number of causal SNPs, we simulate 10 genotype data sets, which results in a total of 40 genotype data sets. With the simulated genotypes and causal effect sizes, we then generated synthetic phenotypes using the two generative models (details below). For each simulated genotype data set, phenotypes are derived under both generative models and for each disease prevalence considered. We consider four prevalences, three methods, and with and without case ascertainment.

And the following to the first paragraph of the simulation results

The following simulation results are based on 10 replications for each parameter setup.

And figure 2 caption we added:

The simulation results are based on 10 replications.

(16) In Figures S4-S7, the authors presented result for "Power ratio", but they did not explain what it means. Therefore, these Figures do not provide any additional information. Moreover, it is difficult to understand the titles of these Figures, in particular their meaning.

We have removed the power ratio from the simulations completely, as they did not properly convey the intended meaning. Instead, we have now expanded on the average power between methods in the simulation results. In particular, we added a power comparison under the proportional hazards model and with case ascertainment. It was originally left out because not all methods identified genome-wide significant SNPs in all parameter setups. We wanted to perform a paired comparison, as the underlying simulated genotypes are the same for each iteration (thereby making them dependent), but it is not possible if no genome-wide significant SNPs are identified in at least one iteration (we can't divide by 0). Instead, we used a pooled approach, ignoring the iteration-wise dependency, but allowing for the comparison to be made. We also mention that the comparison is made on fewer iterations, because not all methods are able to identify genome-wide significant SNPs. The follow was added to the simulation results:

Under the proportional hazards model and with case ascertainment, we were not able to use a paired power for comparisons, as SPACox did not identify any genome-wide significant SNPs in 36 (6 iterations for the ADuLT phenotype and case-control status) out of the 80 iterations performed. The following comparisons are based on the average un-paired power. Under the proportional hazards model, with case ascertainment, and 1000 causal SNPs, ADuLT had a 317% higher power than SPACox and case-control status had a power 256% higher. With 1000 causal SNPs, it is worth noting that SPACox identified 1 or more genome-wide significant SNPs in 11 iterations, while ADuLT and case-control status managed it in 34 iterations. When we only had 250 causal SNPs, ADuLT had a 234% higher power and case-control status had a 193% higher power compared to SPACox.

(17) Figure 3 is misleading. The authors refer to "bars" in the caption, but there are no any bar displayed in this figure. Moreover, abbreviation "M" was not defined.

We believe that we have now addressed this comment in our answer to question 3.

(18) In section 3.2, the authors refer to "the average χ^2 -statistics", but they did not provide any explanation about its meaning, and why in Figures S2-S3 and S6-S7 they refer to "average null χ^2 ". What is the difference?

Average null χ^2 is used in connection to the simulation results. In simulations it is common terminology to use “null” to refer to simulated SNPs with an effect size of 0 (i.e. under the null hypothesis with no effect on the phenotype).

(19) In the same section the authors use term "the null SNPs". They did not provide any information about its meaning. It is a jargon, which were used in a few publications, but which was not accepted by broader scientific community.

“The null SNPs” is used in connection to the simulation results. In simulations it is common terminology to use “null” to refer to simulated SNPs with an effect size of 0 (i.e. under the null hypothesis with no effect on the phenotype). We added the following to the manuscript:

The null SNPs are the SNPs with a simulated effect size of 0 (i.e. no effect). The expected average of these SNPs' χ^2 -statistics is 1.

(20) In a sentence before subsection 3.2.1, the authors wrote:

"In Table S1, simulation results for all parameter setups are available, including the power, relative power compared to SPACox, and mean chi-square statistic of null SNPs."

Could the authors specify where the reader can find "relative power" and "mean chi-square statistic of null SNPs" in Table S1?

We have opted to remove the relative power, as we have decided it is not informative enough (cf. comment #16). However, the mean chi-square statistic of null SNPs has the name “mean_null_chisq” in the table, as the full name is too cumbersome for such a table.

Reviewer #2 (Remarks to the Author):

This paper introduces ADuLT, a new version of a liability model for GWAS data, and compares its performance with Cox regression and logistic regression, focusing particularly on the setting of case-control or case-cohort data where cases are over-represented relative to the population compared to controls. I found the description of the new method somewhat confusing, in part because it seems to be described partly in the methods section and again at the start of the results section. I think I get the basic idea, however: namely, that each case is assigned a “liability threshold” based on the cumulative normal distribution for their age, sex, year of birth (and potentially other variables) using population cumulative incidence rates, and this value is used as a continuous phenotype (for cases, or one minus that for controls); this is then used as the outcome variable in a standard regression model with the genotypes as covariates. Although in the simulation section, it is clear that that entire vector of genotypes, multiplied by their respective effect sizes, is used as the predictor, it isn't clear in the analysis whether each gene is analyzed separately or some multivariate model (which obviously would require some kind of penalization).

We agree that the method description can be improved. An ADuLT GWAS consists of two steps. First, for each individual we estimate their posterior genetic liability conditional on their age or age-of-onset (for cases), sex, and birth-year. To illustrate these posterior genetic liabilities we performed a small example simulation (shown below), where the posterior genetic liabilities are plotted against the true ones. Second, we use the individual genetic liabilities as phenotypes in a genome-wide association study, where each genetic variant is tested (marginally) for association with the genetic liabilities.

Estimated vs True Genetic Liability

We have now made several modifications and additions to the description of the method with the aim to make it easier to understand. Some of these also address comments made by reviewer 1. To improve the method description we have extended the overview of method and the methods section with the following changes:

In the overview of method we have:

- We have improved the description of the ADuLT model (see comment 13 by reviewer 1).

In the methods section we have:

- We highlighted the two step nature of a GWAS with the ADuLT phenotype and the freedom of choice for GWAS software.
- We have formalised how CIPs are used to acquire the personalised threshold.
 - The CIPs are expressed as a function of a person's sex and birth year as well as the current age.
 - The notation introduced here has been used throughout the manuscript.
- Presented an example of how the personalised threshold is used to determine the interval of possible liabilities.
- We highlighted the possibility of fixing the full liability of a case and what it means for the interval of possible liabilities.
- We highlight that the genetic liability (also referred to as the ADuLT phenotype) is estimated based on phenotypic information and does **not** require any genotype information.

The example plot above has now been added to the supplementary notes and a reference has been added in the methods section.

My bigger concern has to do with the implementation of the Cox model. For unsampled data, this would be straightforward, although computationally intensive if the full maximum likelihood or Wald test is used, but less so if the score test (which doesn't require iteration) is used; each gene could be analyzed separately after adjusting for a common set of covariates. (worth explaining that as age is the time scale, including it as a covariate would be unnecessary!). The same computational efficiency by using score tests would apply to logistic regression. What confuses me is the application to sampled case-control data. The appropriate analysis for nested case-control data would then be conditional logistic regression, comparing each case to its sampled risk set members; for case-cohort data, it would be a modified Cox regression comparing each case to the surviving subcohort members, not including the other cases that weren't part of the subcohort, with an appropriate adjustment to the variance estimator. For unmatched case-control data, the appropriate analysis would be unconditional logistic regression, including age as a covariate, but Cox regression without any allowance for the sampling would not be appropriate anyway, hence its poor performance in the sampled simulation settings. So a clearer description of how Cox regression is used is needed.

We agree that the statistical approach must depend on how the data is sampled. As Reviewer #2 says, risk set sampled data, full cohort sampled, and case-cohort sampled data will all lead to the (weighted) partial likelihood function (doi:10.1214/aos/1176324322). In the case of case-cohort data, reviewer #2 says that "it would be a modified Cox regression comparing each case to the surviving subcohort members, not including the other cases that weren't part of the subcohort, ...". This is an option for time-varying samples weights (basically inverse probability weights), as explained by Barlow et al (doi:10.1016/s0895-4356(99)00102-x, table 2). However, Borgan et al (doi:10.1023/a:1009661900674) have shown that an estimator based on time-fixed weights is practically as efficient as analogous estimators based on time-dependent weights. As explained by reviewer #2, Borgan et al also shows how the variance should be adjusted. As long as the design allows a re-weighting back to the target population, martingale residuals

can be obtained as long as the sampling scheme is known. SPACox is simply based on martingale residuals, but unfortunately, SPACox cannot deal with sampled data, (<https://doi.org/10.1016/j.ajhg.2020.06.003>) but only with full cohort data. It is not unusual that data ascertainment or sampling is ignored in time-to-event GWAS, for instance Bi and colleagues (<https://doi.org/10.1016/j.ajhg.2020.06.003>) applied SPACox to UK Biobank data, where some phenotypes are subject to collider bias (<https://doi.org/10.1038/s41467-020-19478-2>) and immortal time bias ([doi:10.1001/jama.2020.9151](https://doi.org/10.1001/jama.2020.9151)) due to how individuals were recruited. Nevertheless, our goal is to compare existing GWAS methods of time-to-event analysis with the LTM-based approach and found a substantial difference between the two types of model when case ascertainment was present in the data, and conventional corrections for the Cox PH model did not resolve the difference.

For time-to-event studies, where researchers are not able to design the study, one has to consider how participants have been sampled/recruited. Depending on the sampling procedure, there are several potential biases present in the data and certain models may not be suitable. For instance, phenotypes in iPSYCH have been ascertained, and UKBB phenotypes are subject to collider bias and immortal time bias [[doi:10.1001/jama.2020.9151](https://doi.org/10.1001/jama.2020.9151) and [doi: 10.1016/j.ajhg.2015.12.019](https://doi.org/10.1016/j.ajhg.2015.12.019)]. The appropriate model for either iPSYCH or UKBB is not a straightforward application of the proportional hazards model without properly considering and accounting for case ascertainment, collider bias, or immortal time bias. Instead a method tailored to each biobank and their particular sampling strategy as they each have different challenges. However, these biobank-specific challenges are often ignored in favour of a one-size-fits-all approach. With the sex and birth year stratified population-representative CIPs, ADuLT is able to account for these biases.

We added the following to the “GWAS in iPSYCH” section to explain the choice of covariates in the GWAS:

For SPACox, we did not include age as a covariate as age was the timeline used. The ADuLT phenotype inherently accounts for age, which means it is not necessary to include as a covariate.

In addition, we added a section to the methods on the proportional hazards model and how it is used in GWAS.

Minor points

Reference [1] is missing authors. Reference [11] is missing the journal.

The references have been updated. Thank you for pointing this out.

P 4 sec 2.2: explain that each genotype is sampled independently, without considering LD. Also in the middle of the second paragraph, the sentence “This model does not account for time-to-event.” Is misleading, as age is part of the individualized liability threshold (via CIP(age)). Ditto lines 3-4 from the end of sec. 2.4.

It is true that we do not model LD, and as such, we added the following line to the simulation details:

Each SNP is simulated independently, which means we do not consider linkage disequilibrium in these simulations, in order to have a clear separation of causal and null SNPs.

We have removed mentions of not being able to account for time-to-event, as it was misleading / confusing (as you pointed out). Instead, we only mention if a method is able to account for time-to-event / is a survival model.

P 5 line 7 up: Eq.[1] hasn't appeared yet.

Middle of p. 13: change neglectable to negligible. Further down, "the least ascertained sample" seems misleading, as I think what you mean is the highest prevalence sample, so if all cases are included, largest proportion of controls are sampled.

We updated the "least ascertained sample" sentence to better reflect our meaning. The full sentence now reads:

However, when considering ascertainment of cases, we found SPACox to have the lowest power of all considered methods under both generative models and for all prevalences except one (the least ascertained sample, i.e. the case ascertained data most similar to the full population).

P 14: the sentence "the assumption that early onset cases have higher disease liability may not always be true", seems to conflate two separate concepts. Obviously as the CIP increases with age, the marginal liability threshold decreases. In a proportional hazards model, the genetic HR is assumed to be constant, but for some diseases (e.g., breast cancer), the genetic HR is larger at younger ages.

The statement is one of the underlying assumptions (or consequences) of the ADuLT model. We essentially assume that the full liabilities are ordered based on the age-of-onset, such that earlier age-of-onset has a higher liability threshold (corresponding to a smaller prevalence) than a later onset. Note that we do not make any statement about the Cox PH model's hazard rate, only about the full liabilities in the ADuLT model.

Reviewer #3 (Remarks to the Author):

General

The authors presented ADuLT, an age-dependent liability threshold model for GWAS. Through both simulation and analyses of data from iPSYCH the authors have demonstrated that ADuLT is more powerful, robust and faster to run than Cox regression models, especially when cases are strongly ascertained (as in iPSYCH). This paper presents an interesting and useful addition to the current GWAS methods, which I think will be very useful for the field going forward.

We appreciate the very kind words, thank you.

Major

1. As the authors described in the introduction, Methods and Results, ADuLT calculates a personalised liability threshold for case-control status inference for all individuals using population-based cumulative incidence proportions (CIPs) for the phenotype of interest as a function of age, sex and birth year. For iPSYCH, the authors derived CIPs from the Danish population registers. CIPs seem only possible to calculate if one has population based data on disease incidence by age; the authors mentioned in various points in the paper that this is becoming more widely available in datasets linked to electronic health records and registers (i.e. independent on the GWAS data). I wonder if the authors thought about generating these CIPs using GWAS data alone (i.e. age of individuals in the GWAS), or using non-population matched CIPs (i.e. using Danish CIPs on UKBiobank for example). It would be ideal to see one analysis to show what would happen in each case, to demonstrate the problems and considerations for each of these non-standard applications of ADuLT. In addition, a brief discussion would be very helpful for users who want to apply this model but do not have population matched CIP.

We believe the CIPs estimated from the GWAS data would not match the true liabilities of the individuals, but they would instead be either over- or underestimated. This bias may be problematic, but the simulations performed in our previous LT-FH++ paper [<https://doi.org/10.1016/j.ajhg.2022.01.009>] suggested a high degree of robustness to misspecification of all considered parameters. One of the considered parameters was the life-time prevalence. It is therefore conceivable that the in-sample CIP could work, especially if sample ascertainment is not severe. This would likely be the easiest way to acquire CIPs if population CIP is not available or unattainable. For the same reason, we believe CIPs obtained from a different population than the target GWAS population could be used. As long as the considered disorders are similar between populations, then the CIPs should be transferable. However, in such a case, we would caution against fixing the full liability of a case, but rather only setting the lower limit to be the individualised threshold and letting the upper limit be infinite.

To illustrate the point on biased liability estimates, consider the classic liability threshold model. Here liabilities will be normally distributed in the entire population, however, in a GWAS sample, it is very rare to have a population-representative (sub)sample. With oversampled cases, we would see a bimodal distribution of liabilities in the resulting GWAS population, but the estimated CIPs would still assume a normal distribution. For undersampled cases, we would have a skewed distribution. We cannot say with certainty that the biased liability estimates would be problematic for GWAS, but we would err on the side of caution and stick to the population representative CIPs until it has been examined further.

Sample ascertainment

Figure: An example of the distribution of liabilities in a non-population representative GWAS sample. Both under- and oversampling of cases is shown.

In the LT-FH++ paper, where we first introduced the ADuLT model, we mention using non-population matched CIPs. We have now added this point to the discussion here as well for completion's sake. We added the following to the discussion:

If population-representative CIPs are not available, it is possible to use CIPs from a similar population. An example of this could be using population-representative CIPs from the Danish registers in UKBB or FinnGen. However, in such a case, we would caution against fixing the full liability of a case, but rather only setting the lower limit to be the individualised threshold and letting the upper limit be infinite.

2. The authors performed GWAS on ADHD, SCZ, ASD and MDD in iPSYCH using ADuLT, and compared its results to GWAS performed using SPACox and Case-control logistic regression, concluding that ADuLT gives more hits (20 independent GWAS hits rather than 17 in case-control and 8 in SPACox). The authors also gave some instances of ADHD hits found only in ADuLT that has been previously reported for ADHD. I would like to see forest plots of the 20 hits found across the 4 diseases for their effects in external datasets (like PGC, UKBiobank, FinnGen etc) to know that across the board the hits obtained by ADuLT are credible. It would be great to compare the replication rate of ADuLT hits vs those found in SPACox and case-control.

As SPACox does not provide effect sizes, we also opted to plot p-values since they are the only way to compare between GWAS methods. We used the largest available summary statistics from PGC[doi: 10.1038/s41588-022-01285-8, doi: 10.1038/s41588-019-0344-8, doi: 10.1038/s41593-018-0326-7, doi: 10.1038/s41586-022-04434-5] for the four considered phenotypes, used the SNPs that were present in the iPSYCH data, and performed LD clumping and prioritising the SNPs lowest p-values from the meta-analysed PGC summary statistics to keep only independent hits. Then, we took the SNPs that were genome-wide

significant in the LD-clumped PGC summary statistics and plotted the p-values of the three considered methods against each other. In short, the plot below shows the p-values for ADuLT against case-control status and SPACox p-values on the SNPs that are independent genome-wide significant from the largest available PGC summary statistics, and they are therefore not necessarily genome-wide significant for either of the methods considered.

The red dashed line is the identity line. The blue line is the best fitting line with an intercept at 0. The blue line indicates that p-values are on average the same or slightly more significant for ADuLT than case-control status and SPACox.

3. Why was sex added as covariates in ADuLT GWAS (and age not), when it is (both are) already used to define the liability threshold of each individual? Why should/shouldn't it matter? Please explain. The supplementary materials showed manhattans with and without adding age as covariate, so clearly the authors tried to find out if adding in age as covariate made a difference - did they try this just to be consistent with covariates selected for case-control GWAS? Turns out adding in age or not for ADuLT seems to matter a tiny bit (few hits different), which is not obvious unless I looked really carefully - suggest presenting in a table summarising results with/without age as covariate in addition to manhattans.

Thank you for bringing this up, as it is a really good point. We have gone back and made the analysis where sex is not a covariate in the ADuLT analysis either. We have included the Manhattan plots in the supplemental notes. The most common way to deal with covariates in a GWAS setting is by projecting it out of the model, which for sex, means one sex has a fixed value subtracted from their outcome. For a binary phenotype or quantitative phenotypes such as height, this may be useful, as there are biological effects that can be removed (e.g. mean height differences between men and women). However, it may not be the correct thing to do for the ADuLT phenotype, as the phenotype itself incorporates the effect of sex and age (as you observantly pointed out). When accounting for sex as a covariate, it is a fixed value that will be projected out, regardless of its interaction with age. In the ADuLT phenotype, the threshold is determined by the interaction between sex and age,

then it is not only a simple shift, but a more complex transformation through the CIPs. Therefore, we believe the correct analysis is actually to not include sex or age as a covariate, since those variables will be accounted for in the phenotype through the CIPs. We do not expect there to be any biological mechanisms that could influence sex in the autosome for the analysed phenotypes. We have now performed the GWAS without sex as a covariate, but results remain largely similar. The analysis without sex has been added to the supplementary notes.

We added the following to the discussion:

It is not necessary to include age as a covariate in a GWAS with ADuLT as the outcome. The effect of age is already accounted for in the refined phenotype itself. Traditionally, age or some related variable is included in the analysis to account for a person's lifespan and period of being at-risk. A common way to deal with such covariates in a regression is to project them out and consider only the univariate regression with the regressed outcome and predictor. Therefore, projecting out the covariates boils down to subtracting a value from the observations. This subtraction is not necessary with the ADuLT phenotype, as the effect of age and sex can be accounted for through the sex and birth year stratified CIPs. In fact, accounting for sex and age through the CIPs provide a more detailed way of accounting for this information, as interactions are also considered. We have also performed the GWAS analyses without sex and age as covariates for the ADuLT phenotype, but it did not differ substantially from when excluding them.

We have also added a table with the number of significant SNPs across all iPSYCH disorders for each method with and without sex and age as covariates. The Manhattan plots for the ADuLT GWAS without sex and age as covariates are also added to the supplementary notes. The table is shown here.

Method	with age	with sex	total significant SNPs
ADuLT	no	no	20
	no	yes	20
	yes	yes	17
CaseControl	no	no	16
	no	yes	16
	yes	yes	17
SPACox	no	yes	8
	yes	yes	14

Table S3: Table including the sum of genome-wide significant associations across the four iPSYCH disorders.

4. Simulations and its associated figures (figure 2, bunch of supplementary figures): the down-sampling experiment was to examine the effect of ascertainment of cases - why not just call it "ascertained", to be crystal clear? I had to tell myself downsampling really means ascertainment when I read figure 2 and all similar supplementary figures, as downsampling alone (if non-ascertained) is not going to result in power differences we see in those figures, ascertainment would.

We have had a lot of internal discussion about what to call it going back to our previous paper on LT-FH++. At some point, we landed on calling it downsampling, since that is what we actually do to achieve the case ascertainment. However, after having read the manuscript again, we agree that calling it case ascertainment is more proper here. As you highlight, downsampling does not necessarily result in case ascertainment. In the simulation details, we will add a statement on how the case ascertainment is achieved in the simulations, but we will refer to it as a case ascertained sample, rather than a downsampled sample.

Proportional hazards models have previously been proposed to analyse time-to-event phenotypes in genome-wide association studies (GWAS). While proportional hazards models have many useful applications, their ability to identify genetic associations under different generative models where ascertainment is present in the analysed data is poorly understood.

This includes widely used study designs such as case-control and case-cohort designs (e.g. the iPSYCH study design) where cases are commonly ascertained.

Here we examine how recently proposed and computationally efficient Cox regression for GWAS perform under different generative models with and without ascertainment. We also propose the age-dependent liability threshold model (ADuLT), first introduced as the underlying model for the LT-FH++ method, as an alternative approach for time-to-event GWAS. We then benchmark ADuLT with SPACox and standard case-control GWAS using simulated data with varying degrees of ascertainment. We find Cox regression GWAS to underperform when cases are strongly ascertained (cases are oversampled by a factor larger

than 5), regardless of the generative model used. In contrast, we found ADuLT to be robust to case-control ascertainment, while being much faster to run. We then used the methods to conduct GWAS for four psychiatric disorders, ADHD, Autism, Depression, and Schizophrenia in the iPSYCH case-cohort sample, which has a strong case-ascertainment. Summarising across all four mental disorders, ADuLT found 20 independent genome-wide significant associations, while case-control GWAS found 17 and SPACox found 8, consistent with our simulation results.

As more genetic data are being linked to electronic health records, robust GWAS methods that can make use of age-of-onset information have the opportunity to increase power in analyses for common health outcomes. We find ADuLT to be a computationally efficient and robust time-to-event GWAS method that, in terms of statistical power, performs on par with or better than Cox-regression GWAS, both in simulations and real data analyses of four psychiatric disorders. ADuLT has been implemented in an R package called LTFHPlus, and is available on GitHub.

REVIEWERS' COMMENTS

Reviewer #1 (Remarks to the Author):

In the revised version, the authors considered a major part of suggestions proposed by the reviewers. Related modifications improved the manuscript considerably and made it more transparent and solid.

I have only one comment and a few minor suggestions, which are outlined below and do not affect the conclusions of the manuscript.

(1) Possible contradictory statements:

The authors added the following paragraph:

"Under the proportional hazards model and with case ascertainment, we were not able to use a paired power for comparisons, as SPACox did not identify any genome-wide significant SNPs in 36 (6 iterations for the ADuLT phenotype and case-control status) out of the 80 iterations performed. The following comparisons are based on the average un-paired power. Under the proportional hazards model, with case ascertainment, and 1000 causal SNPs, ADuLT had a 317% higher power than SPACox and case-control status had a power 256% higher. With 1000 causal SNPs, it is worth noting that SPACox identified 1 or more genome-wide significant SNPs in 11 iterations, while ADuLT and case-control status managed it in 34 iterations. When we only had 250 causal SNPs, ADuLT had a 234% higher power and case-control status had a 193% higher power compared to SPACox. Plots of the power as a function of MAF and the true effect size can be found in Figures S8 to S15."

I was confused with this paragraph. There are two contradictory statements.

According to the first one, the ADuLT and case-control approaches are better than SPACox one:

"Under the proportional hazards model, with case ascertainment, and 1000 causal SNPs, ADuLT had a 317% higher power than SPACox and case-control status had a power 256% higher."

According to the second statement, SPACox is better than the ADuLT and and case-control because the former approach can detect significant (causal) SNPs with probability better than 1/11, while the latter ones can detect any significant SNP with probability 1/34: "With 1000 causal SNPs, it is worth noting that SPACox identified 1 or more genome-wide significant SNPs in 11 iterations, while ADuLT and case-control status managed it in 34 iterations."

Could the authors rewrite this paragraph making the statements more transparent and consistent?

=====

Minor suggestions

=====

(2) References:

There are pairs of identical references: Refs.20 and 21, Refs. 23 and 24.

References are not ordered as they were cited in the text.

Could the authors check the Reference List and make corrections if they are needed?

(3) Figures:

Figure numbering is not in order of their appearance in the main text.

Reviewer #2 (Remarks to the Author):

The authors have been very responsive to my original comments. I've also reviewed their responses to the other two reviewers, which also strike me as very detailed and generally satisfactory, although I defer judgement to those reviewers. In particular, I now appreciate that the inclusion of Cox regression as a comparison approach is intended to show that it isn't really an appropriate method for ascertained samples (as reflected in its poorer performance) and the clearer discussion of why it's inappropriate, despite it's frequent use in GWAS.

The only additional comment I'd like to make at this point (which should have been obvious previously, but didn't occur to me then) is the following. The marginal distribution of liabilities in case-control samples is highly bimodal, as illustrated in the first figures in their responses to reviewer 2 (myself) and 3. This would appear to violate one of the basic assumptions of linear regression, that the residuals should be normally distributed with constant variance. Of course, it's well known that linear regression is quite robust to violation of these assumptions, I wonder if it isn't worth a brief comment to that, since neither logistic nor Cox regression (which condition on the outcome variable) makes such assumptions. As this is a minor concern, I don't feel strongly about a response being needed, so defer to the authors' (and editor's) judgment.

Reviewer #3 (Remarks to the Author):

I am satisfied with the authors replies to my comments and appreciate the work they have done answering other reviewers' requests too. I recommend acceptance.

REVIEWER COMMENTS

This colour is used for reviewer comments.

This colour is used for replies to reviewer comments.

Changes to the text have been marked in red.

Reviewer #1 (Remarks to the Author):

In the revised version, the authors considered a major part of suggestions proposed by the reviewers. Related modifications improved the manuscript considerably and made it more transparent and solid.

We thank the reviewer for appreciating the hard work we put in improving our manuscript based on the previous reviewers' comments.

I have only one comment and a few minor suggestions, which are outlined below and do not affect the conclusions of the manuscript.

(1) Possible contradictory statements:

The authors added the following paragraph:

"Under the proportional hazards model and with case ascertainment, we were not able to use a paired power for comparisons, as SPACox did not identify any genome-wide significant SNPs in 36 (6 iterations for the ADuLT phenotype and case-control status) out of the 80 iterations performed. The following comparisons are based on the average un-paired power. Under the proportional hazards model, with case ascertainment, and 1000 causal SNPs, ADuLT had a 317% higher power than SPACox and case-control status had a power 256% higher. With 1000 causal SNPs, it is worth noting that SPACox identified 1 or more genome-wide significant SNPs in 11 iterations, while ADuLT and case-control status managed it in 34 iterations. When we only had 250 causal SNPs, ADuLT had a 234% higher power and case-control status had a 193% higher power compared to SPACox. Plots of the power as a function of MAF and the true effect size can be found in Figures S8 to S15."

I was confused with this paragraph. There are two contradictory statements.

According to the first one, the ADuLT and case-control approaches are better than SPACox one:

"Under the proportional hazards model, with case ascertainment, and 1000 causal SNPs, ADuLT had a 317% higher power than SPACox and case-control status had a power 256% higher."

According to the second statement, SPACox is better than the ADuLT and and case-control because the former approach can detect significant (causal) SNPs with probability better than 1/11, while the latter ones can detect any significant SNP with probability 1/34: "With 1000 causal SNPs, it is worth noting that SPACox identified 1 or more genome-wide significant SNPs in 11 iterations, while ADuLT and case-control status managed it in 34 iterations."

Could the authors rewrite this paragraph making the statements more transparent and consistent?

We are sorry about the confusion in the wording. We meant that the method found at least 1 significant variant in 11 iterations *out of the 40 simulations*. To avoid this confusion, and improve clarity, we have rephrased the paragraph as follows:

Under the proportional hazards model and with case ascertainment, we were not able to use a paired power for comparisons, as SPACox was unable to identify any genome-wide significant SNPs in 36 out of 80 simulations (29 out of 40 when simulating 1000 causal SNPs, and 7 out of 40 when simulating 250 causal SNPs). The ADuLT phenotype and case-control status were unable to identify genome-wide significant SNPs in only 6 out of 80 simulations, all of which are for 1000 causal SNPs. The following comparisons are based on the average un-paired power. Under the proportional hazards model simulations with case ascertainment and 1000 causal SNPs, ADuLT had a 317% higher power than SPACox, whereas case-control status had 256% higher power than SPACox. When simulating 250 causal SNPs, ADuLT resulted in a 234% higher power and case-control status had a 193% higher power compared to SPACox.

We note that instead of reporting that SPACox found one or more genome-wide significant variants in 11 iterations (out of 40), we now say that it did not find any genome-wide significant variants in 29 out of 40.

=====
Minor suggestions
=====

(2) References:

There are pairs of identical references: Refs.20 and 21, Refs. 23 and 24.

References are not ordered as they were cited in the text.

Could the authors check the Reference List and make corrections if they are needed?

Thank you for noticing. It has been solved.

(3) Figures:

Figure numbering is not in order of their appearance in the main text.

In the publish-ready formatting, it should now be resolved. Thank you.

Reviewer #2 (Remarks to the Author):

The authors have been very responsive to my original comments. I've also reviewed their responses to the other two reviewers, which also strike me as very detailed and generally satisfactory, although I defer judgement to those reviewers. In particular, I now appreciate that the inclusion of Cox regression as a comparison approach is intended to show that it isn't really an appropriate method for ascertained samples (as reflected in its poorer performance) and the clearer discussion of why it's inappropriate, despite its frequent use in GWAS.

We thank the reviewer for appreciating the hard work we put in improving our manuscript based on the previous reviewers' comments.

The only additional comment I'd like to make at this point (which should have been obvious previously, but didn't occur to me then) is the following. The marginal distribution of liabilities in case-control samples is highly bimodal, as illustrated in the first figures in their responses to reviewer 2 (myself) and 3. This would appear to violate one of the basic assumptions of linear regression, that the residuals should be normally distributed with constant variance. Of course, it's well known that linear regression is quite robust to violation of these assumptions, I wonder if it isn't worth a brief comment to that, since neither logistic nor Cox regression (which condition on the outcome variable) makes such assumptions. As this is a minor concern, I don't feel strongly about a response being needed, so defer to the authors' (and editor's) judgment.

You are correct. The ADuLT phenotype is bimodal, and this can cause issues when the prevalence of the phenotype is very low. We have now added a comment in the discussion that some care should be given if the phenotype has a low in-sample prevalence. Notably, it has not been a problem for the analysis we performed here (due to ascertainment bias in the IPSYCH cohort). We refer to the work done in connection to REGENIE by Mbatchou et al. (<https://doi.org/10.1038/s41588-021-00870-7>), where they recommend not using BOLT-LMM (a linear mixed model) when the in-sample prevalence is lower than 1/80. In connection to a different project, we have done some investigation of this problem and we believe the underlying problem is similar for the ADuLT phenotype as what is reported by the REGENIE authors, and therefore we recommend a similar minimum in-sample prevalence as suggested by the REGENIE authors. An alternative solution to this problem is applying a rank-based inverse normal transformation (McCaw et al., *Biometrics* 2020).

Across the GWAS sample, the ADuLT phenotype is often bimodal in practice. The bimodality is due to the underlying truncated normal distributions leading to a gap between the resulting mean genetic estimates of the cases and controls. It has been shown that binary phenotypes when analysed with linear mixed models can suffer from inflation when the in-sample prevalence is low (Zhou et al., *Nat Genet* 2020; Mbatchou et al., *Nat Genet* 2021). The ADuLT phenotype is quantitative, and is therefore not suitable for logistic regression (such as SAIGE or REGENIE), although it is often bimodal and may result in a clear separation between cases and controls. A potential solution is to employ rank-based inverse normal transformation (RINT) to the ADuLT phenotype (McCaw et al., *Biometrics* 2020), but it may lead to a loss in power. Therefore, we recommend not using the ADuLT phenotype

for GWAS when the in-sample prevalence is lower than roughly 1/80, which is in line with the advice provided by Mbatchou et al. for the application of mixed linear models to binary outcomes.

Reviewer #3 (Remarks to the Author):

I am satisfied with the authors replies to my comments and appreciate the work they have done answering other reviewers' requests too. I recommend acceptance.

Thank you, we appreciate the recommendation.